# VIDEO-THINKER: SPARKING "THINKING WITH VIDEOS" VIA REINFORCEMENT LEARNING

## ABSTRACT

Recent advances in image reasoning methods, particularly "Thinking with Images", have demonstrated remarkable success in Multimodal Large Language Models (MLLMs); however, this dynamic reasoning paradigm has not yet been extended to video reasoning tasks. In this paper, we propose Video-Thinker, which empowers MLLMs to think with videos by autonomously leveraging their intrinsic "grounding" and "captioning" capabilities to generate reasoning clues throughout the inference process. To spark this capability, we construct Video-Thinker-10K, a curated dataset featuring autonomous tool usage within chain-of-thought reasoning sequences. Our training strategy begins with Supervised Fine-Tuning (SFT) to learn the reasoning format, followed by Group Relative Policy Optimization (GRPO) to strengthen this reasoning capability. Through this approach, Video-Thinker enables MLLMs to autonomously navigate grounding and captioning tasks for video reasoning, eliminating the need for constructing and calling external tools. Extensive experiments demonstrate that Video-Thinker achieves significant performance gains on both in-domain tasks and challenging out-of-domain video reasoning benchmarks, including Video-Holmes, CG-Bench-Reasoning, and VR-Bench. Our Video-Thinker-7B substantially outperforms existing baselines such as Video-R1 and establishes state-of-the-art performance among 7B-sized MLLMs.

## 1 INTRODUCTION

Multimodal Large Language Models (MLLMs) have embraced a revolutionary paradigm shift toward "Thinking with Images" for image understanding and reasoning tasks, evolving from passively treating images as static context to actively localizing, zooming in, and reasoning over image content during the thinking process (Zheng et al., 2025; Liu et al., 2024b; Shen et al., 2024; Wang et al., 2025e; Ma et al., 2024). This dynamic multimodal reasoning paradigm has yielded substantial advances on MLLMs across diverse image reasoning tasks, including visual question answering (Liu et al., 2023; Zhao et al., 2025; Gupta & Kembhavi,

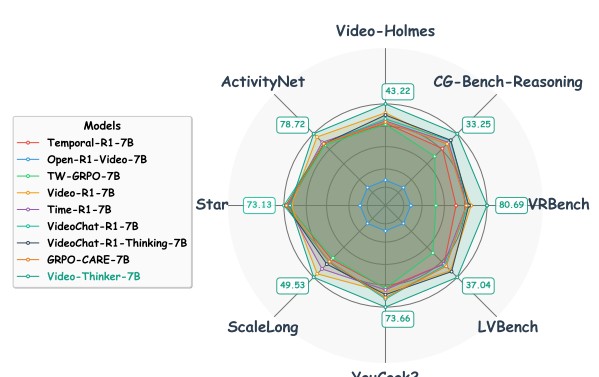

Figure 1: Overall Performance of Video-Thinker

2023; Liu et al., 2024c), visual mathematical problem solving (Chen et al., 2025c; Shao et al., 2024a; Wang et al., 2025a; Yue et al., 2024; Li et al., 2025a), and complex scene understanding (You et al., 2023; Yang et al., 2023; Zhang et al., 2025b; Zheng et al., 2025; Ma et al., 2025b). However, the extension of these capabilities to video understanding presents significant challenges. Unlike static images, videos inherently contain temporal dependencies, motion patterns, and evolving visual narratives that require sophisticated temporal reasoning mechanisms, whereas MLLMs struggle to dynamically manipulate and reason over temporal sequences without relying on explicitly pre-designed chain-of-thought prompting strategies (Fei et al., 2024; Feng et al., 2025; Shi et al., 2024).

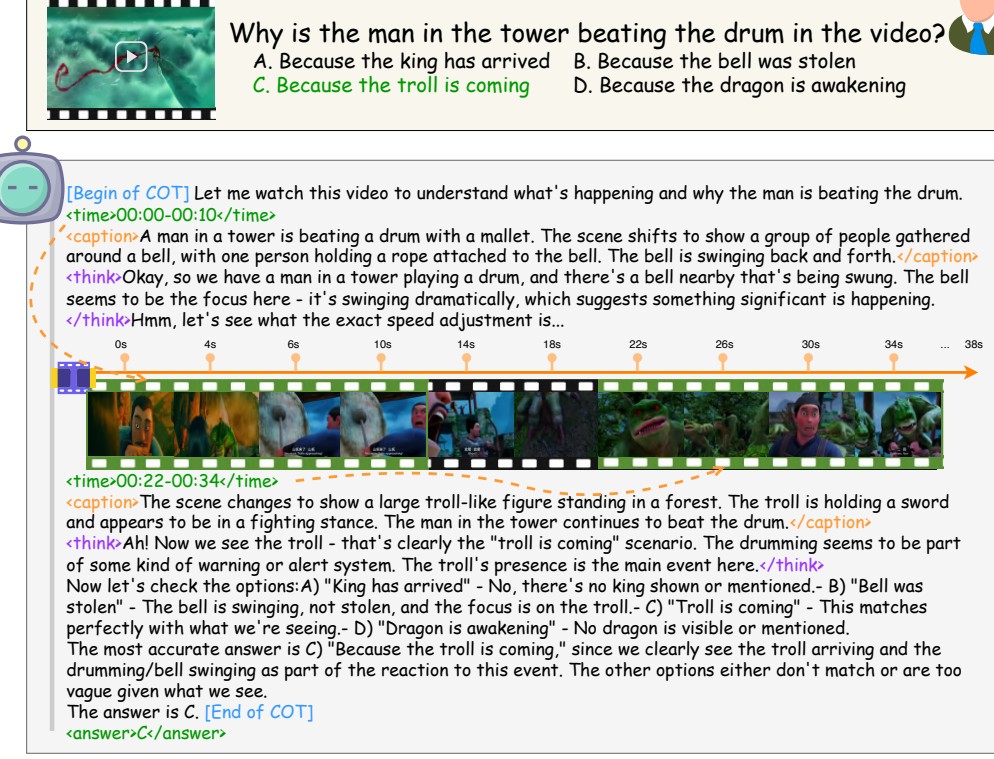

Figure 2: Video-Thinker integrates "grounding" and "captioning" capabilities throughout the reasoning process using end-to-end reinforcement learning.

In this paper, we propose a novel framework named **Video-Thinker** to enhance MLLMs by enabling them to perform visual reasoning through structured video analysis capabilities. Drawing inspiration from spatial visual operations in "Thinking with Images" (OpenAI, 2024) for image understanding — such as "crop" for region localization and "zoom-in" for detailed region comprehension — we introduce the following temporal visual operations - namely "grounding" and "captioning". The "grounding" operation serves as a temporal localization mechanism that identifies and extracts key frames containing critical visual information within the video sequence, while the "captioning" operation functions as a comprehension mechanism that analyzes these key frames to extract, interpret, and synthesize relevant visual cues into a coherent understanding. Fortunately, these video localization and comprehension capabilities can be developed within MLLMs themselves, thereby eliminating the need for MLLMs to adapt to and invoke external handcrafted tools. Hence, our Video-Thinker can enable structured temporal reasoning through chain-of-thought (CoT) processes, allowing models to autonomously navigate and analyze specific temporal segments rather than treating videos as monolithic inputs. The framework orchestrates these temporal manipulation capabilities through systematic reasoning traces that synthesize visual cues across multiple video segments. Our approach differs fundamentally from previous investigations in two key aspects. First, unlike video-of-thoughts methodologies that rely on sophisticated pre-designed CoT processes (Fei et al., 2024), our framework develops intrinsic temporal reasoning capabilities that emerge naturally from the training process. Second, in contrast to general visual reasoning models that require extensive datasets exceeding 160K samples (Feng et al., 2025), our approach demonstrates that effective video reasoning capabilities can be achieved with significantly greater efficiency using only 10K carefully curated training examples.

To instantiate our framework, we carefully construct **Video-Thinker-10K**, a curated training dataset of 10K samples spanning diverse video-reasoning tasks and domains. Each sample comprises strategically selected key video segments, detailed captions describing visual clues for each temporal window, and structured reasoning traces that demonstrate how to synthesize these multimodal cues for complex video understanding tasks. As illustrated in Figure 2, our reasoning trace adopts a structured format wherein each key video segment is systematically processed through three specialized annotation tags: the `<time></time>` tag for precise temporal localization, the `<caption></caption>` tag for comprehensive visual cue extraction, and the `<think></think>` tag for analytical reasoning that synthesizes the extracted visual information.

Our training methodology employs a two-stage approach: we first conduct supervised fine-tuning (SFT) using our curated thought processes as ground truth supervision to establish foundational format-following capabilities. We subsequently apply Group Relative Policy Optimization (GRPO) (Shao et al., 2024b) for reinforcement learning, where only the final answer serves as the outcome reward. This approach enables the model to intrinsically acquire both grounding and captioning capabilities, facilitating autonomous temporal navigation for sophisticated video reasoning tasks. Our extensive experiments demonstrate that Video-Thinker achieves the state-of-the-art (SOTA) performance among 7B-sized MLLMs across various challenging out-of-domain video reasoning benchmarks, including Video-Holmes (Cheng et al., 2025), CG-Bench-Reasoning Chen et al. (2024a), and VRBench (Yu et al., 2025b), as demonstrated in Figure 1.

While emerging concurrent work (Liu et al., 2025a) focuses on developing external tools to enhance video understanding capabilities, our approach takes a fundamentally different direction. Video-Thinker introduces a novel paradigm of "Thinking with Videos" by intrinsically integrating video grounding and captioning capabilities directly within the CoT reasoning process, eliminating the need for external tool dependencies. Our comprehensive evaluation demonstrates the effectiveness of this approach at the 7B parameter scale, showing competitive or superior performance compared to tool-use methods or approaches relying on powerful external grounding and captioning models. This work seeks to inspire discussion and provide insights into an important question: whether to enhance model capabilities through intrinsic training (SFT and RL) or to integrate external specialized tools for specific functionalities.

Our main contributions are summarized as follows: (i) proposing a new paradigm (Video-Thinker) of "Thinking with Videos" by intrinsically integrating grounding and captioning capabilities within the CoT process, eliminating the dependency on external tools; (ii) contributing a meticulously curated video reasoning dataset (Video-Thinker-10K) encompassing comprehensive localization annotations and rich comprehension information; and (iii) empirically setting new SOTA performances across multiple video reasoning benchmarks.

## 2 RELATED WORK

Recent advances in reinforcement learning-based post-training have demonstrated significant improvements in reasoning capabilities, as evidenced by OpenAI-o1 (Jaech et al., 2024) and Deepseek-R1 (Guo et al., 2025b). Building upon this foundation, the field of MLLMs is undergoing a paradigmatic shift in how visual information is integrated into reasoning processes. Traditionally, MLLMs have treated images as static inputs, relegating the reasoning process entirely to the textual domain (Su et al., 2025). An emerging paradigm, however, elevates visual information to an explicit, manipulable intermediate within the reasoning process itself, transforming vision from a passive input into an active cognitive tool (OpenAI, 2024). This approach is exemplified by several recent works: Deepeyes (Zheng et al., 2025) employs end-to-end reinforcement learning to train models that autonomously invoke visual tools (e.g., magnification) while interleaving visual and textual CoT reasoning, effectively enabling models to "Think with Images". Visual-ARFT (Liu et al., 2025b) utilizes GRPO (Shao et al., 2024b) to develop capabilities in task planning, stepwise reasoning, and tool use, allowing models to strategically employ Python-based image-processing operators.

The natural extension of these advances lies in video reasoning, which represents a core capability for MLLMs seeking to capture the logical structure of temporal visual content—a crucial step beyond mere video perception toward genuine video understanding (Wang & Peng, 2025; Dang et al., 2025; Yu et al., 2025a). Recent efforts have begun addressing this challenge: Video-R1 (Feng et al., 2025) extends GRPO into the video domain, promoting implicit temporal reasoning alongside spatial reasoning capabilities. VideoChat-R1 (Li et al., 2025d) leverages reinforcement fine-tuning to strengthen spatiotemporal localization while preserving conversational proficiency. Temporal-R1 (Li et al., 2025c) employs explicit temporal grounding rewards and variance-aware data selection strategies to enhance both semantic and temporal reasoning with improved data efficiency.

Despite these advances, current approaches remain largely confined to either temporal localization or standalone video reasoning, falling short of integrating temporal grounding seamlessly into the CoT processes. Our proposed Video-Thinker framework — extending the paradigm of "Think with Images" — enables MLLMs to "Think with Videos" by facilitating dynamic navigation of temporal content within the reasoning process. Specifically, Video-Thinker incorporates "grounding" and "captioning"

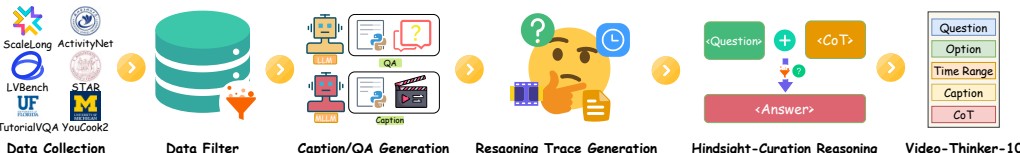

Figure 3: Data synthesis pipeline of Video-Thinker-10K where the data distribution is depicted in Figure 5 in Appendix B.

capabilities as integral components of the CoT reasoning, allowing MLLMs to systematically attend to, interpret, and analyze relevant temporal segments throughout video-based tasks.

# 3 THINK WITH VIDEOS: FROM DATA SYNTHESIS TO MODEL TRAINING

As video reasoning tasks require temporal localization and comprehension capabilities in MLLMs, we propose "grounding" and "captioning" as fundamental anchors for model enhancement. To address this requirement, we first establish high-quality curated data termed Video-Thinker-10K, using a new hindsight-curation reasoning method, as detailed in Section 3.1. Subsequently, we train our Video-Thinker models on these datasets through supervised fine-tuning and reinforcement learning approaches, as described in Section 3.2.

## 3.1 DATA SYNTHESIS VIA HINDSIGHT-CURATION REASONING

Here, we curate a diverse collection of source data from the following six prominent datasets, namely ActivityNet (Caba Heilbron et al., 2015), TutorialVQA (Colas et al., 2019), YouCook2 (Zhou et al., 2018b), STAR (Wu et al., 2024), ScaleLong (Ma et al., 2025a), and LVBench (Wang et al., 2024). These sources span a wide spectrum of domains — ranging from human activities and instructional tutorials to cooking procedures, situated reasoning, and long-form content such as TV series. Within these datasets, we identified the following two complementary categories of data: (i) Caption-labeled datasets, including ActivityNet, TutorialVQA, and YouCook2, provide detailed, human-annotated captions for specific temporal intervals within key video segments but lack complex questions that require deep reasoning capabilities. (ii) QA-labeled datasets, comprising STAR, ScaleLong, and LVBench, offer challenging question-answer pairs designed for deep reasoning but lack the granular, per-segment visual descriptions essential for our structured reasoning framework.

To inspire MLLMs with intrinsic capabilities for "grounding" and "captioning", our training data curation is guided by two core principles. One is: our training data requires questions that compel MLLMs to localize multiple key segments, accurately summarize their content, and synthesize this information to derive comprehensive answers. The other one is: our training data must provide supervision through a structured reasoning trace that includes the `<time></time>` tag for temporal localization, the `<caption></caption>` tag for visual cue description, and the `<think></think>` tag for analytical reasoning, explicitly integrating temporal actions within the CoT process. To bridge the gap between the collected source data and the expected structured data samples described above, we developed a systematic data transformation pipeline, as demonstrated in Figure 3).

We first applied quality filters to remove corrupted videos and exclude videos with fewer than 64 frames to ensure adequate temporal content. Our pipeline then branches into two distinct generation strategies based on dataset characteristics: (i) For caption-labeled datasets (namely, ActivityNet, TutorialVQA, YouCook2) that are rich in temporal annotations and segment descriptions, we focused on synthesizing corresponding reasoning questions. We leveraged DeepSeek-R1 (Guo et al., 2025a) to generate complex multiple-choice questions that necessitate reasoning across multiple video segments, using the existing detailed segment descriptions as the contextual foundation. (ii) For QA-labeled datasets (namely, STAR, ScaleLong, LVBench) that provide high-quality question-answer pairs but lack granular per-segment descriptions, we concentrated on generating the missing visual cues. Given the ground-truth answers and temporal annotations, we employed Gemini-2.5-Flash-Lite (Comanici et al., 2025) to produce answer-conditioned descriptive captions for video segments, ensuring that the generated visual descriptions are relevant to the reasoning process.

Finally, with both question-answer pairs and segment-level visual descriptions now available across all data samples, we perform the final reasoning trace synthesis. We use DeepSeek-V3 (Liu et al.,

2024a) for reverse-curation generation, where the model receives the ground-truth answer, generated visual descriptions (captions), and temporal annotations to produce high-quality reasoning processes that articulate step-by-step temporal analysis. Each trace adheres to our predefined structured format, incorporating the `<time></time>` tag for temporal localization, the `<caption></caption>` tag for visual evidence summarization, and the `<think></think>` tag for analytical reasoning elaboration, thereby creating complete training instances for our Video-Thinker-10K dataset.

To ensure that the generated "grounding" and "captioning" components are beneficial for the final response, previous data synthesis pipelines such as Video-Holmes (Cheng et al., 2025) employ manual sampling inspection to ensure quality and relevance. To reduce the cost of human evaluation and annotation, we propose a novel hindsight curation process. For each sample, the generated content within the `<time></time>` and `<caption></caption>` tags is input into Qwen2.5-VL-7B-Instruct (Bai et al., 2025) to evaluate whether the model can derive the correct answer. If the model fails to produce the accurate answer, we regenerate the reasoning trace. This iterative process repeats up to three times, ensuring that all samples are equipped with a high-quality and relevant reasoning trace that effectively guides the model toward the correct solution. Also, we carefully sample from these sources to ensure a balanced distribution across various tasks and domains, as detailed in Figure 5 in Appendix B. We also provide the specific prompt templates used in this generation pipeline in Appendix D.

## 3.2 Training Strategy of Video-Thinker

Let $D = (V, Q, T, Y) \in \mathcal{D}_{\text{Video-Thinker}}$ denote any sample in Video-Thinker-10K constructed in the above subsection, where $V$ represents the video, $Q$ is the question, $T$ is the ground-truth reasoning trace containing grounding and captioning contents, and $Y$ is the ground-truth answer.

**SFT Optimization for Format-Following.** We start by Supervised Fine-tuning (SFT) to bootstrap Video-Thinker's ability to generate structured reasoning traces over "grounding" and "captioning" contents. Since pre-trained MLLMs lack exposure to our specialized reasoning format with `<time></time>`, `<caption></caption>`, and `<think></think>` tags, SFT provides essential cold-start initialization by teaching the model to follow high-quality reasoning patterns from our Video-Thinker-10K dataset.

Formally, the SFT objective is to minimize the negative log-likelihood of the target reasoning trace $T$ and final answer $Y$, where the loss function can be formulated as:

$$\mathcal{L}_{\text{SFT}}(\theta) = -\mathbb{E}_{(V,Q,Y) \sim \mathcal{D}_{\text{Video-Thinker}}} \left[ \sum_{t=1}^{|[T;Y]|} \log p_\theta \left( [T;Y]_t \middle| V, Q, [T;Y]_{<t} \right) \right], \quad (1)$$

where $[T; Y]$ denotes the concatenation of $T$ and $Y$, and $p_\theta$ is the policy of Video-Thinker model parameterized by $\theta$. Namely, the model is trained to predict each subsequent token $[T; Y]_t$ of the reasoning trace and the final answer, conditioned on the video $V$, the question $Q$, and the preceding tokens $[T; Y]_{<t}$.

**GRPO Optimization for Autonomous Navigation over Grounding and Captioning Capabilities.** To achieve sophisticated video reasoning with autonomous navigation over grounding and captioning capabilities, we employ Group Relative Policy Optimization (GRPO) to further optimize Video-Thinker beyond the above SFT stage. GRPO eliminates the need for value function approximation by generating multiple candidate responses for each $(V, Q, Y)$ sample and assessing their relative quality through verifiable rewards. Formally, for each $(V, Q, Y)$ sampled from $\mathcal{D}_{\text{Video-Thinker}}$, GRPO generates $G$ distinct reasoning traces $\{T^{(1)}, T^{(2)}, \ldots, T^{(G)}\}$ using the current policy $p_{\theta_{\text{old}}}$. The policy is optimized by maximizing:

$$\mathcal{J}_{\text{GRPO}}(\theta) = \mathbb{E}_{(V,Q,T,Y) \sim \mathcal{D}_{\text{Video-Thinker}}} \left[ \frac{1}{G} \sum_{i=1}^{G} \left( \min \left( \frac{\pi_\theta}{\pi_{\theta_{\text{old}}}} A_i, \quad \text{clip}\left( \frac{\pi_\theta}{\pi_{\theta_{\text{old}}}}, 1 - \epsilon, 1 + \epsilon \right) A_i \right) \right. \right.$$
$$\left. \left. - \beta \, \text{KL}\left( p_\theta(\cdot|V,Q) \middle\| p_{\text{ref}}(\cdot|V,Q) \right) \right) \right], \quad (2)$$

where $\pi_\theta = p_\theta(T^{(i)}|V,Q)$, $\pi_{\theta_{\text{old}}} = p_{\theta_{\text{old}}}(T^{(i)}|V,Q)$, $\text{KL}(p_\theta(\cdot|V,Q)\|p_{\text{ref}}(\cdot|V,Q))$ denotes the KL divergence (Van Erven & Harremos, 2014) between the current policy $p_\theta(\cdot|V,Q)$ and reference

Table 1: Comparison of model performance on video reasoning datasets in both in-domain and out-of-domain settings. The best results are marked in **red bold** and the second best in blue.

| Model | Out of Domain | | | | | | In Domain | | | | |
|---|---|---|---|---|---|---|---|---|---|---|---|
| | Video-Holmes | CG-Bench-Reasoning | VRBench | SciVideoBench | VdeoTT | VideoMME | ActivityNet | Star | ScaleLong | YouCook2 | LVBench |
| *Open-source Vanilla Models* | | | | | | | | | | | |
| InternVL-2.5-8B | 20.52% | 19.39% | 26.74% | 15.50% | 26.62% | 29.89% | 45.52% | 49.85% | 26.81% | 40.84% | 23.91% |
| InternVL-3-8B | 18.67% | 24.23% | 41.14% | 20.50% | 28.42% | 39.93% | 48.56% | 51.34% | 29.34% | 51.15% | 25.93% |
| Qwen2.5-VL-7B-Instruct | 34.02% | 27.10% | 63.42% | 21.40% | 34.57% | 51.85% | 70.96% | 69.25% | 40.06% | 63.74% | 33.33% |
| Qwen2.5-Omni-7B | 29.99% | 23.85% | 49.04% | 16.80% | 36.27% | 45.44% | 63.92% | 59.40% | 36.91% | 54.58% | 31.65% |
| MiMo-VL-7B-RL-2508 | 28.96% | 26.16% | 57.94% | 16.70% | 34.15% | 45.70% | 49.84% | 48.06% | 35.33% | 44.27% | 28.62% |
| *Open-source Reasoning Models* | | | | | | | | | | | |
| Temporal-R1-7B | 33.81% | 25.27% | 60.92% | 20.80% | 35.31% | 51.63% | 70.88% | 70.15% | 39.75% | 63.74% | 32.66% |
| Open-R1-Video-7B | 21.83% | 16.46% | 50.15% | 16.90% | 18.66% | 37.41% | 55.76% | 44.48% | 31.86% | 50.76% | 26.94% |
| TW-GRPO-7B | 33.32% | 22.11% | 53.46% | 20.50% | 33.93% | 48.30% | 70.00% | 71.04% | 39.12% | 63.74% | 29.97% |
| Video-R1-7B | 38.54% | 27.81% | 69.25% | 23.90% | 41.04% | 54.81% | 76.00% | 67.76% | 47.32% | 65.65% | 34.68% |
| Time-R1-7B | 34.73% | 28.28% | 66.48% | 21.00% | 34.78% | 53.59% | 72.00% | 70.44% | 44.47% | 64.50% | 32.65% |
| VideoChat-R1-7B | 35.65% | 29.26% | 67.65% | 22.80% | 35.63% | 54.41% | 70.88% | 73.13% | 40.69% | 69.08% | 32.99% |
| VideoChat-R1-Thinking-7B | 37.45% | 29.44% | 67.81% | 20.30% | 35.95% | 54.15% | 70.88% | 71.64% | 41.95% | 66.79% | 35.01% |
| GRPO-CARE-7B | 34.34% | 27.49% | 66.39% | 21.30% | 35.74% | 54.22% | 70.96% | 71.34% | 40.69% | 68.32% | 33.33% |
| VersaVid-R1-7B | 37.07% | 28.58% | 67.72% | 21.80% | 35.31% | 54.78% | 72.32% | 71.94% | 40.69% | 66.79% | 34.34% |
| VideoRFT-7B | 24.39% | 23.77% | 61.54% | 19.60% | 37.65% | 47.85% | 50.72% | 43.58% | 36.59% | 58.40% | 26.94% |
| VR-Thinker-7B | 25.37% | 19.54% | 53.43% | 21.20% | 31.07% | 43.56% | 55.36% | 63.58% | 32.18% | 51.91% | 30.98% |
| Video-RTS-7B | 29.56% | 18.09% | 27.71% | 20.30% | 35.10% | 42.63% | 63.60% | 65.07% | 30.28% | 65.27% | 20.88% |
| *SFT Model* | | | | | | | | | | | |
| Video-Thinker-SFT-7B | 31.52% | 24.95% | 62.40% | 16.90% | 33.93% | 46.26% | 70.80% | 64.18% | 43.22% | 56.11% | 35.69% |
| *Pure RL Model* | | | | | | | | | | | |
| Video-Thinker-Pure-RL-7B | 30.70% | 22.47% | 63.06% | 23.70% | 36.37% | 45.52% | 64.24% | 46.57% | 41.96% | 45.04% | 30.64% |
| *Our Model* | | | | | | | | | | | |
| Video-Thinker-7B | **43.22%** | **33.25%** | **80.69%** | **26.30%** | **42.42%** | **54.96%** | **78.72%** | 70.66% | **49.53%** | **73.66%** | **37.04%** |

policy $p_{\text{ref}}(\cdot|V, Q))$, $A_i$ is the advantage for the $i$-th reasoning trace, and $\epsilon$ and $\beta$ are hyperparameters Here, the advantage $A_i$ is computed using outcome supervision based on normalized rewards within each group. Specifically, for each reasoning trace $T^{(i)}$, we assign a reward $r^{(i)}$ comprising both correctness and format components:

$$r^{(i)} = r^{(i)}_{\text{correct}} + r^{(i)}_{\text{format}}, \tag{3}$$

where $r^{(i)}_{\text{correct}} \in \{0, 1\}$ indicates whether the extracted answer from reasoning trace $T^{(i)}$ matches the ground truth $Y$, and $r^{(i)}_{\text{format}}$ measures adherence to the structured reasoning format with `<time></time>`, `<caption></caption>`, and `<think></think>` tags. The advantages are then computed as:

$$A_i = \tilde{r}^{(i)} = \frac{r^{(i)} - \text{mean}(\{r^{(j)}\}_{j=1}^{G})}{\text{std}(\{r^{(j)}\}_{j=1}^{G})} \tag{4}$$

This approach enables the model to learn from relative comparisons within each group, promoting both accurate reasoning and proper temporal structure adherence.

**Aha Moment.** We find that Video-Thinker demonstrates the capacity for complex reasoning through self-reflective behaviors, which can be characterized as "aha moments" (Guo et al., 2025a). The model exhibits metacognitive processes by periodically revisiting its initial interpretations of video grounding and captioning tasks, critically evaluating and refining its outputs when necessary. This self-corrective behavior suggests that Video-Thinker transcends simple pattern matching and instead engages in dynamic internal feedback mechanisms similar to Video-R1 (Feng et al., 2025), while requiring substantially less training data (10K compared to 160K samples). This phenomenon is illustrated in Figure 4, with additional examples provided in Appendix G.

## 4 EXPERIMENT

### 4.1 EXPERIMENTAL SETUP

**Datasets and Benchmarks.** To comprehensively assess the video reasoning performance of Video-Thinker, we conduct evaluations under both in-domain and out-of-domain settings. For the in-domain evaluation, since the TutorialVQA (Colas et al., 2019) training set contains only 76 samples, we do not

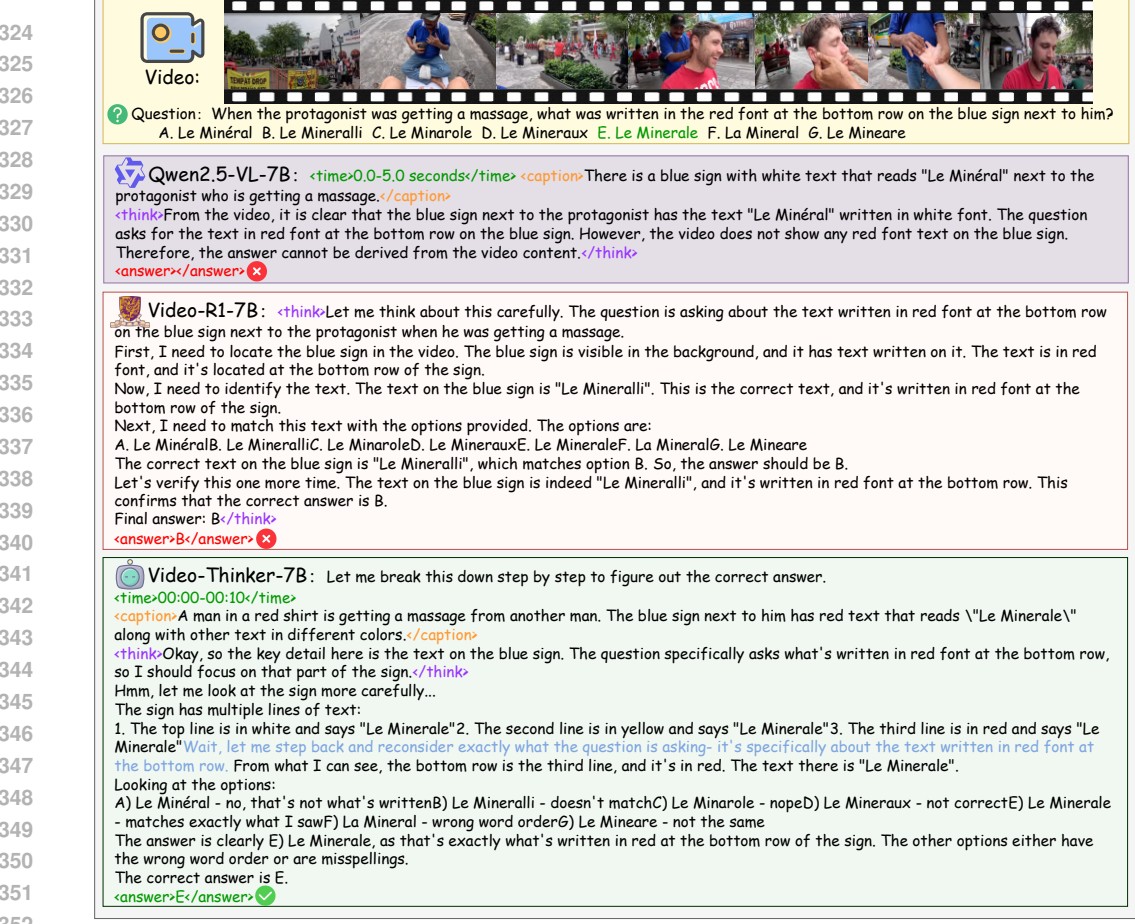

Figure 4: An example of Video-Thinker-7B's reasoning output on CG-Bench-Reasoning dataset.

construct a corresponding test set. Instead, we derive held-out test sets from the five training datasets - ActivityNet (Caba Heilbron et al., 2015), LVBench (Wang et al., 2024), ScaleLong (Ma et al., 2025a), Star (Wu et al., 2024), and YouCook2 (Zhou et al., 2018a) - by splitting them at a ratio of 1:9 between test and training subsets. For the out-of-domain evaluation, we select six datasets featuring complex video reasoning tasks: Video-Holmes (Cheng et al., 2025), CG-Bench-Reasoning (Chen et al., 2024a), VRBench (Yu et al., 2025b), SciVideoBench (Deng et al., 2025), Video-TT (Zhang et al., 2025a), and VideoMME (Fu et al., 2024).

**Baseline Models.** To comprehensively evaluate the effectiveness of Video-Thinker, we conduct extensive comparisons against two distinct categories of baseline models: (i) five open-source vanilla models, including InternVL-2.5-8B (Chen et al., 2024b), InternVL-3-8B (Zhu et al., 2025), Qwen2.5-VL-7B-Instruct (Bai et al., 2025), Qwen2.5-Omni-7B (Xu et al., 2025), and MiMo-VL-RL-2508 (Xiaomi, 2025); and (ii) twelve open-source reasoning models, comprising Temporal-R1-7B (Li et al., 2025c), Open-R1-Video-7B (Wang & Peng, 2025), TW-GRPO-7B (Dang et al., 2025), Video-R1-7B (Feng et al., 2025), Time-R1-7B (Wang et al., 2025d), VideoChat-R1-7B (Li et al., 2025d), VideoChat-R1-Thinking-7B (Li et al., 2025d), GRPO-CARE-7B (Chen et al., 2025d), VersaVid-R1-7B (Chen et al., 2025b), VideoRFT-7B (Wang et al., 2025b), VR-Thinker-7B (Wang et al., 2025c), and Video-RTS-7B (Wang et al., 2025f).

**Training Details.** We employ Qwen2.5-VL-7B-Instruct (Bai et al., 2025) as our base model. During the SFT stage, we train the model on our Video-Thinker-10K dataset for 1 epoch using a learning rate of $1 \times 10^{-5}$ and a batch size of 16. For the subsequent GRPO stage, we set the hyperparameter $\beta$ in the KL divergence term to 0.04. To ensure training stability, we apply a weight decay rate of 0.01 and clip the maximum gradient norm to 5. The initial learning rate is configured to $5 \times 10^{-6}$ with a batch size of 8. Both training stages utilize the same prompt template, as detailed in Appendix D. For computational efficiency during both training phases, we subsample each video to a maximum of 16 frames and process each frame at a maximum resolution of $128 \times 28 \times 28$ pixels.

Table 2: Comparison of model performance on video reasoning datasets with different numbers of frames during inference in both in-domain and out-of-domain settings. The best results are marked in **red bold** and the second best in blue.

| Model | # Frames | Out of Domain | | | In Domain | | | | |
|---|---|---|---|---|---|---|---|---|---|
| | | Video-Holmes | CG-Bench-Reasoning | VRBench | ActivityNet | Star | ScaleLong | YouCook2 | LVBench |
| Qwen2.5-VL-7B-Instruct | 16 | 34.02% | 27.10% | 63.42% | 70.96% | 69.25% | 40.06% | 63.74% | 33.33% |
| | 32 | 34.89% | 30.33% | 64.45% | 73.36% | 71.04% | 43.53% | 64.89% | 36.36% |
| | 64 | 37.56% | 32.16% | 65.91% | 74.40% | **74.03%** | 45.18% | 68.32% | **39.39%** |
| Video-R1-7B | 16 | 38.54% | 27.81% | 69.25% | 76.00% | 67.76% | 47.32% | 65.65% | 34.68% |
| | 32 | 40.56% | 29.29% | 69.44% | 77.20% | 70.15% | 49.84% | 66.03% | 37.37% |
| | 64 | 40.94% | 30.12% | 70.23% | 77.76% | 72.54% | 50.26% | 66.79% | 37.04% |
| Video-Thinker-7B | 16 | 43.22% | 33.25% | 80.69% | 78.72% | 70.66% | 49.53% | 73.66% | 37.04% |
| | 32 | 43.39% | 33.88% | 80.91% | **79.68%** | 72.24% | 51.74% | 74.05% | 38.38% |
| | 64 | **44.15%** | **35.59%** | **81.29%** | 78.96% | 72.24% | **52.04%** | **74.05%** | 37.71% |

## 4.2 Performance Comparisons and Analysis

We evaluate all baseline models on the aforementioned dataset using accuracy as the primary evaluation metric. The performance of our Video-Thinker-7B compared to various baseline methods is summarized in Table 4. The results yield the following key findings.

**Video-Thinker-7B achieves a new SOTA performance on video reasoning benchmarks among 7B-sized MLLMs.** As demonstrated in Table 1, our proposed Video-Thinker-7B establishes new SOTA results both in-domain and out-of-domain settings across various video reasoning benchmarks. The model demonstrates particularly strong performance on challenging out-of-domain tasks, achieving 43.22% on Video-Holmes (a 4.68% improvement over the best baseline), 33.25% on CG-Bench-Reasoning (3.81% improvement over the best baseline), and 80.69% on VRBench (11.44% improvement over the best baseline). These substantial improvements validate the effectiveness of our Video-Thinker framework in inspiring MLLM's "grounding" and "captioning" capabilities over video sequences. We also observe that employing either SFT or RL in isolation fails to improve the performance of Video-Thinker-RL-7B. Our results demonstrate that effective training requires a sequential approach: SFT first establishes format-following capabilities, while subsequent RL training enhances autonomous reasoning abilities.

**GRPO stage yields substantial improvements in MLLM out-of-domain generalization over SFT stage.** A critical finding from our experimental analysis is that GRPO training performance substantially outperforms that of SFT in terms of video reasoning generalization. The GRPO-trained Video-Thinker-7B demonstrates marked superiority over its SFT counterpart, with improvements of 11.70% on Video-Holmes (43.22% vs. 31.52%), 8.30% on CG-Bench-Reasoning (33.25% vs. 24.95%), and 18.29% on VRBench (80.69% vs. 62.40%). These gains are particularly pronounced in out-of-domain evaluation scenarios. Importantly, Video-Thinker-SFT-7B consistently underperforms relative to most baseline methods and even degrades below the base model Qwen2.5-VL-7B-Instruct across several benchmarks, revealing the limited generalization capacity of SFT alone. Nevertheless, SFT serves an essential role in enabling the model to acquire our structured reasoning format. These findings establish the necessity of a two-stage training paradigm: initial SFT stage for format acquisition, followed by GRPO stage for data-efficient performance enhancement and robust cross-domain generalization.

**Video-Thinker-7B constantly outperforms the baseline methods with different numbers of video frames during inference.** To investigate the impact of video frame count on model performance, we evaluate Video-Thinker-7B against two baseline models, Qwen2.5-VL-7B and Video-R1-7B, using 16, 32, and 64 frames during inference across all in-domain and out-of-domain settings. As presented in Table 2, several key observations emerge from this analysis. First, increasing the number of input frames consistently enhances performance across most benchmarks and all evaluated models, with 64 frames yielding optimal results in the majority of cases. This trend suggests that richer temporal information enables more comprehensive video understanding and reasoning. Second, Video-Thinker-7B consistently outperforms both baseline models across all tested frame counts, demonstrating superior capability in processing and integrating temporal information. The performance gap between Video-Thinker-7B and the baselines remains substantial regardless of frame count, indicating that our model's performance improvements for video reasoning are effective across different temporal sampling strategies.

Table 3: Comparison of model performance on video grounding and captioning tasks. The best results are marked in **red bold** and the second best in blue.

| Model | Grounding | | | | Captioning | | | |
|---|---|---|---|---|---|---|---|---|
| | **mIOU** | **Recall@0.3** | **Recall@0.5** | **Average** | **Meteor** | **ROUGE-L** | **BLEU@1** | **Average** |
| Qwen2.5-VL-7B | 27.47 | 39.52 | 23.71 | 30.23 | 14.10 | 14.91 | 10.15 | 13.05 |
| Video-R1-7B | – | – | – | – | 12.72 | 11.64 | 7.52 | 10.63 |
| Video-Thinker-7B | **48.22** | **79.29** | **51.49** | **59.67** | **15.87** | **20.11** | **15.34** | **17.11** |

In addition to analyzing the impact of video frame count, we also present the performance of Video-Thinker-7B under varying training steps and learning rates during the GRPO stage in Appendix F.

## 4.3 IN-DEPTH ANALYSIS OF GROUNDING AND CAPTIONING CAPABILITIES

One of the main ideas underlying Video-Thinker is that "grounding" and "captioning" capabilities serve as key "tools" for video reasoning. Therefore, we further investigate whether the performance gains of Video-Thinker stem from enhanced grounding and captioning capabilities. To validate the improved temporal manipulation capabilities of Video-Thinker, we conduct quantitative experiments to analyze the "grounding" and "captioning" abilities of Video-Thinker-7B, comparing it against the base model Qwen2.5-VL-7B-Instruct and the previous SOTA model Video-R1-7B. For both experiments, we select 1K samples from caption-labeled in-domain test dataset with ground truth caption annotations and temporal annotations (sourced from ActivityNet (Caba Heilbron et al., 2015), YouCook2 (Zhou et al., 2018a), and TutorialVQA (Colas et al., 2019)). Each sample contains one or multiple ground truth question-relevant key segment time annotations for grounding ability verification and corresponding ground truth captions for captioning ability evaluation.

**Video-Thinker-7B demonstrates superior performance across all evaluated metrics in video grounding tasks.** To assess temporal grounding capabilities, we employ a structured evaluation protocol wherein models are prompted to answer questions while simultaneously outputting question-relevant time segments within `<time></time>` tags (detailed prompt specifications provided in Appendix D). We subsequently extract model-predicted temporal segments and evaluate their alignment with ground truth annotations using two complementary metrics: mean Intersection-over-Union (mIoU) and Recall@K.

As demonstrated in Table 3, Video-Thinker-7B consistently outperforms baseline models across all evaluation metrics. Our model achieves an mIoU of 48.22%, representing a substantial 75.5% improvement over Qwen2.5-VL-7B's 27.47%. For recall metrics, Video-Thinker-7B attains 79.29% and 51.49% for Recall@0.3 and Recall@0.5, respectively, nearly doubling the baseline performance (39.52% and 23.71%). The overall averaged performance of 59.67% constitutes a 97% relative improvement compared to the baseline's 30.23%. Note that Video-R1 is excluded from this evaluation due to its inability to follow our prompt to generate temporal annotations within our templates.

**Video-Thinker-7B demonstrates superior performance across all evaluated metrics in video captioning tasks.** To evaluate captioning capabilities, we prompt models to generate descriptions for video segments using the instruction "Describe the video segment", then compare predicted captions against ground truth references. We employ three established metrics: BLEU@1 (Papineni et al., 2002), METEOR (Banerjee & Lavie, 2005), and ROUGE-L (Lin, 2004).

The captioning results presented in Table 3 demonstrate that Video-Thinker-7B achieves superior performance across all three evaluation metrics. Specifically, our model attains 15.87% METEOR, 20.11% ROUGE-L, and 15.34% BLEU@1, yielding an overall average of 17.11%. Compared to the base model Qwen2.5-VL-7B-Instruct, Video-Thinker exhibits consistent improvements of 1.77%, 5.20%, and 5.19%, respectively, representing a 31.2% relative enhancement in overall performance. When compared against Video-R1-7B, the improvements are even more pronounced, with gains of 3.15%, 8.47%, and 7.82% respectively, achieving a 61.0% relative improvement in overall performance. These results substantiate Video-Thinker's enhanced capacity for generating contextually accurate and temporally relevant video descriptions.

Moreover, to further validate the importance of grounding and captioning capabilities for video understanding, we conduct additional experiments by providing ground-truth grounding and captioning annotations to Video-R1-7B and evaluating its performance on the Video-Holmes (Cheng et al.,

Table 4: Performance comparison of our model with video reasoning methods with external tool use. The best results are marked in **red bold** and the second best in blue.

| Model | Video-Holmes | CG-Bench-Reasoning | VRBench |
|---|---|---|---|
| *Base Model* | | | |
| Qwen2.5-VL-7B-Instruct | 34.02% | 27.10% | 63.42% |
| *Base Model + Plug-and-play Tools* | | | |
| Grounding: Temporal-R1-7B Captioning: SkyCaptioner-V1-8B | 30.58% | 22.80% | 55.09% |
| Grounding: Qwen2.5-VL-7B-Instruct Captioning: Qwen2.5-VL-7B-Instruct | 31.23% | 24.05% | 59.14% |
| Grounding: Qwen2.5-VL-72B-Instruct Captioning: Qwen2.5-VL-72B-Instruct | 33.96% | 25.99% | 60.54% |
| *Existing Tool-use Method* | | | |
| VideoMind-7B | 38.98% | 31.99% | 75.39% |
| *Our Model (Endogenous)* | | | |
| Video-Thinker-7B | **43.22%** | **33.25%** | **80.69%** |

2025). As detailed in Appendix E, these oracle experiments demonstrate that access to accurate video grounding and captioning information significantly enhances MLLM performance.

### 4.4 IN-DEPTH ANALYSIS OF INTERNAL CAPABILITIES AND EXTERNAL TOOL CALLING

One of the main claims in our paper is that our CoT design and data synthesis with hindsight-curation reasoning enhance the captioning and grounding capabilities inherent to our method. To validate this claim, we conduct a comparative analysis examining whether external tool integration can achieve similar performance improvements. We compare our Video-Thinker against several baseline configurations that equip the base model Qwen2.5-VL-7B-Instruct with external specialized tools: the grounding model Temporal-R1-7B (Li et al., 2025b) and the captioning model Skycaptioner-V1-8B (Chen et al., 2025a). Counterintuitively, our results reported in Table 4 demonstrate that incorporating these external tools actually degrades the performance of the base model. We hypothesize that this performance degradation stems from compatibility issues between the specialized models and Qwen2.5-VL-7B-Instruct. To test this hypothesis, we construct an alternative baseline where Qwen2.5-VL-7B itself, and its larger counterpart Qwen2.5-VL-72B, serve as both the grounding and captioning modules. However, as reported in Table 4, this configuration also fails to improve upon the base model's capabilities. This counterintuitive finding warrants deeper investigation. Through the case studies presented in Figure 15 (Appendix G.2), we observe that CoT information from external tools can mislead Qwen2.5-VL-7B-Instruct when the reasoning chain becomes discontinuous. These case studies reveal systematic risk in tool invocation and result integration, suggesting that effective tool utilization requires sophisticated coordination mechanisms to maintain reasoning coherence.

To further validate our approach, we compare Video-Thinker-7B against Video-Mind-7B (Liu et al., 2025a), a model specifically fine-tuned to orchestrate multiple video understanding tools, including planners, grounders, verifiers, and answerers. Our results suggest that well-designed internal reasoning capabilities can outperform external tool integration for models of similar scale.

## 5 CONCLUSION AND FUTURE WORK

In this work, we introduce Video-Thinker, a novel approach that extends the "Thinking with Images" paradigm to video reasoning by empowering MLLMs to autonomously leverage their intrinsic grounding and captioning capabilities. Through the construction of the Video-Thinker-10K dataset and a two-stage training strategy combining SFT and GRPO, our method enables MLLMs to generate reasoning clues throughout the inference process without relying on external tools, and our resulting Video-Thinker-7B model establishes SOTA performance among 7B-sized models. Looking forward, it is interesting to scale Video-Thinker with larger model sizes or with additional intrinsic capabilities beyond grounding and captioning, or with more modalities such as audio.

ETHICS STATEMENT

This work focuses on the study of multimodal video understanding and reasoning. All datasets used in our experiments are publicly available and commonly adopted in prior research. We followed the respective dataset licenses and usage terms. No personally identifiable information (PII) or sensitive private data was collected, generated, or annotated by the authors. Our study does not raise direct ethical concerns such as misuse of personal data, harmful content, or bias amplification beyond what is already inherent in the benchmark datasets. We acknowledge that large-scale vision-language models may inherit biases present in training data. To mitigate risks, our evaluations were restricted to established academic benchmarks for fair comparison. We encourage future researchers and practitioners to be mindful of potential social implications when applying these systems in downstream applications.

REPRODUCIBILITY STATEMENT

In order to ensure reproducibility, we provide a comprehensive description of datasets, model implementations, and experimental settings in the main paper and the appendix. The benchmarks and evaluation metrics we used are standard and publicly available. All baselines are either taken from released model checkpoints or trained/evaluated with publicly accessible open-source implementations. To further promote reproducibility, hyperparameters, training details, and evaluation protocol are clearly documented. We release our code at `https://anonymous.4open.science/status/Video-Thinker-F78A` to enable the community to fully reproduce our results. We commit to following ICLR guidelines for transparency and reproducibility in scientific reporting.

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

## A  OVERALL ALGORITHM OF VIDEO-THINKER

---

**Algorithm 1** Video-Thinker

---

**Input:** Collected dataset $\mathcal{D}_{\text{source}}$ according to Section 3.1, pre-trained MLLM with parameters $\theta$
**Output:** MLLM trained by the Video-Thinker
 1: **Phase 1: Data Synthesis via Hindsight-curation Reasoning according to Section 3.1**
 2: **for** each sample $(V, Q, T, Y) \in \mathcal{D}_{\text{source}}$ **do**
 3:     Generate missing visual captions and reasoning questions.
 4:     Synthesize structured reasoning trace $T$ with hindsight curation as detailed in Section 3.1.
 5: **end for**
 6: Construct Video-Thinker-10K dataset $\mathcal{D}_{\text{Video-Thinker}}$.
 7: **Phase 2: SFT Optimization for Format-Following according to Section 3.2**
 8: **for** each $(V, Q, T, Y) \in \mathcal{D}_{\text{Video-Thinker}}$ **do**
 9:     Compute and minimize: $\mathcal{L}_{\text{SFT}}(\theta)$ according to Eq. (1).
10: **end for**
11: **Phase 3: GRPO Optimization for Autonomous Navigation according to Section 3.2**
12: **for** each $(V, Q, T, Y) \in \mathcal{D}_{\text{Video-Thinker}}$ **do**
13:     Generate $G$ reasoning traces $\{T^{(i)}\}_{i=1}^{G}$ using current policy.
14:     Compute rewards $r^{(i)} = r_{\text{correct}}^{(i)} + r_{\text{format}}^{(i)}$ according to Eq. (3).
15:     Calculate normalized advantages $A_i = \frac{r^{(i)} - \text{mean}(\{r^{(j)}\})}{\text{std}(\{r^{(j)}\})}$ according to Eq. (4).
16:     Optimize GRPO objective $\mathcal{J}_{\text{GRPO}}(\theta)$ with clipped importance sampling according to Eq. (2).
17: **end for**
18: **return** MLLM with tuned $\theta$

---

## B  DATA DISTRIBUTION OVER SOURCE DATASETS IN SECTION 3.1

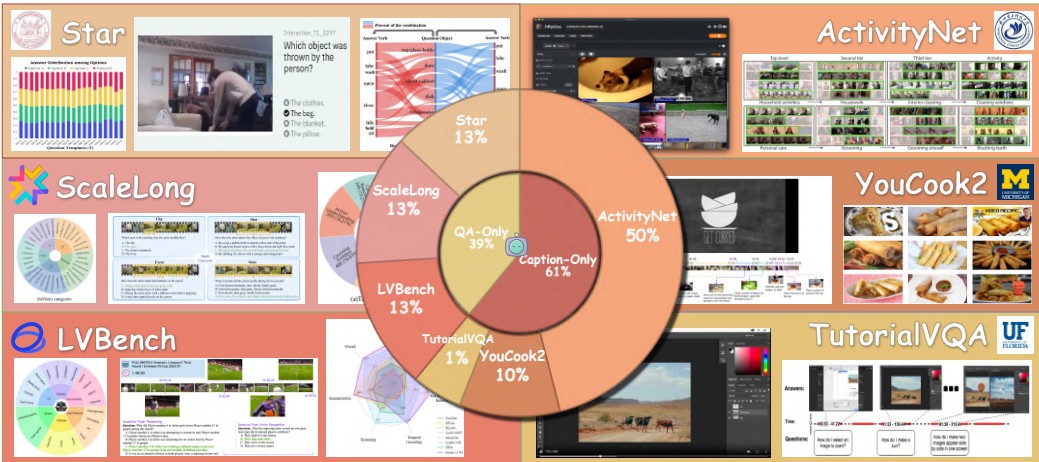

Figure 5: The data distribution of our Video-Thinker-10K dataset.

## C  EXPERIMENT CONFIGURATION

### C.1  DATASETS AND BENCHMARKS

**ActivityNet** (Caba Heilbron et al., 2015) is a large-scale VideoQA benchmark, consisting of 5,800 long untrimmed videos (average length ∼180s) and 58K bilingual (Chinese/English) human-annotated QA pairs. Introducing question templates over motion, spatial and temporal relations as well as free-form queries, offering a robust testbed for spatio-temporal reasoning and fine-grained comprehension.

**STAR** (Wu et al., 2024) focuses on situated reasoning in daily life scenarios, covering 22K short clips and 60K structured questions spanning interaction, sequence, prediction, and feasibility reasoning. Constructing "situational hyper-graphs" to capture entities, actions, and relations, ensuring explicit logical grounding and reducing shortcut biases.

**ScaleLong** (Ma et al., 2025a) targets multi-scale temporal understanding in long videos, with 269 videos (avg. 86 minutes) and 1.7K well-curated QA pairs. Each question is aligned with one of four temporal granularities—clip, shot, event, story—thus isolating evaluation across distinct timescales without conflating video content.

**YouCook2** (Zhou et al., 2018a) contains 2,000 instructional cooking videos from 89 recipes, with temporal annotations and imperative descriptions for stepwise procedures. As a standard benchmark for instructional video understanding, it enables research into activity recognition, weakly supervised object grounding, and cross-video procedural knowledge transfer.

**LVBench** (Wang et al., 2024) evaluates long-horizon multimodal reasoning with 103 YouTube videos (117 total hours) and 1.5K QA pairs. Tasks emphasize summarization, causal reasoning, and temporal localization, with additional "clue-length" annotations specifying the minimal evidence span required.

**Video-Holmes** (Cheng et al., 2025) uniquely probes narrative-driven reasoning via 270 mystery films and 1.8K QA pairs. It emphasizes multi-clue integration, causal inference, and social relation reasoning, filling a crucial gap in evaluating complex video storylines beyond surface perception.

**CG-Bench** (Chen et al., 2024a) consists of 1.2K long videos and 12K QA pairs, introducing a clue-grounded paradigm for perception, reasoning, and hallucination queries. Its white-box and black-box evaluations require explicit evidence retrieval, mitigating guess-based shortcuts and incentivizing faithful video-grounded reasoning. We used the reasoning section of CG-Bench while evaluating.

**VRBench** (Yu et al., 2025b) benchmarks multi-step reasoning over 1,010 narrative videos spanning 8 languages. Providing high-quality stepwise reasoning annotations and a multi-phase evaluation pipeline to jointly assess reasoning process and outcome, is a first benchmark to explicitly measure both the "how" and "what" of video reasoning.

**SciVideoBench** (Deng et al., 2025) focuses on advanced video reasoning within the scientific domain, consisting of 1,000 multiple-choice questions derived from 240+ experimental videos across 25 specialized subjects. Demanding sophisticated domain-specific knowledge and intricate logical reasoning, it addresses the critical gap in evaluating higher-order multimodal cognitive skills beyond general perception.

**Video-TT** (Zhang et al., 2025a) assesses the correctness and robustness of video interpretation, comprising 5,000 question-answer pairs across 1,000 YouTube Shorts. Probing visual and narrative complexity through open-ended and adversarial questions, it aims to quantify the gap between video LLMs and human intelligence in maintaining consistent performance under challenging real-world conditions.

**VideoMME** (Fu et al., 2024) introduces a full-spectrum benchmark for multi-modal video analysis, featuring 2,700 question-answer pairs over 900 videos with durations ranging from 11 seconds to 1 hour. Integrating multi-modal inputs including subtitles and audio with rigorous expert annotations, it provides a comprehensive assessment of MLLM capabilities across varying contextual dynamics and data modalities.

### C.2 BASELINE MODELS

**InternVL-2.5-8B** (Chen et al., 2024b) refines the InternVL architecture with progressive scaling strategies, improved training pipelines, and high-quality data filtering. It achieves competitive results against leading commercial systems, excelling in multi-image/video understanding, document parsing, and multimodal reasoning benchmarks.

**InternVL-3-8B** (Zhu et al., 2025) further enhances perception and reasoning by introducing Native Multimodal Pre-Training, Variable Visual Position Encoding, and Mixed Preference Optimization. Beyond vision-language tasks, it extends capabilities to GUI agents, 3D vision perception, and tool usage, setting new standards for multimodal flexibility.

**Qwen2.5-VL-7B** (Bai et al., 2025) emphasizes long-form video understanding with dynamic temporal modeling and efficient frame-rate training. It supports structured outputs for documents and visual grounding, while also enabling agentic tool-use behaviors across vision and language tasks.

**Qwen2.5-VL-Omni-7B** (Xu et al., 2025) unifies text, image, audio, and video into a novel end-to-end architecture (Thinker-Talker) with real-time speech generation and streaming interaction. Its multimodal coverage allows robust conversational agents that can handle both text and voice outputs.

**Temporal-R1-7B** (Li et al., 2025c) introduces a dual-reward reinforcement learning scheme that balances semantic correctness with temporal localization accuracy. Promoting more robust spatio-temporal reasoning in long video contexts.

**Time-R1-7B** (Wang et al., 2025d) extends beyond retrospective understanding to future event prediction and hypothetical scenario generation. It showcases efficient training curricula for advancing temporal intelligence in MLLMs.

**Open-R1-Video-7B** (Wang & Peng, 2025) and **Video-R1** (Feng et al., 2025) adapt the R1 reinforcement learning paradigm to video reasoning with GRPO-driven optimization. Both emphasize temporal-aware training strategies, achieving strong results on challenging video benchmarks.

**TW-GRPO-7B** (Dang et al., 2025) refines RL pipelines with token-wise weighting and soft reward mechanisms, producing denser and more fine-grained reasoning chains.

**GRPO-CARE-7B** (Chen et al., 2025d) enhances logical consistency using a coherence-aware reward design, improving the alignment between intermediate reasoning steps and final predictions.

**VideoChat-R1-7B** (Li et al., 2025d) integrates structured video reasoning with interactive dialogue, supporting temporally grounded conversation in multimodal applications. It represents a step toward practical, user-facing video reasoning systems.

**VersaVid-R1-7B** (Chen et al., 2025b) addresses the paradigm conflict between divergent captioning and convergent QA tasks in RL-enhanced video MLLMs. Introducing a training framework with intermediate proxy tasks—DarkEventInfer and MixVidQA—it compels models to simultaneously develop holistic understanding and precise reasoning, effectively bridging the gap to achieve significant performance gains in both capabilities.

**VideoRFT-7B** (Wang et al., 2025b) extends the Reinforcement Fine-Tuning (RFT) paradigm to MLLMs to cultivate human-like video reasoning. Utilizing a multi-expert-driven CoT curation pipeline to generate high-quality training data and a novel semantic-consistency reward to align textual reasoning with visual evidence, it achieves state-of-the-art performance across six video reasoning benchmarks.

**MiMo-VL-RL-2508** (Xiaomi, 2025) introduces powerful vision-language models trained via Mixed On-policy Reinforcement Learning (MORL) integrating diverse reward signals. By incorporating high-quality reasoning data into pre-training and optimizing across multi-domain tasks, it delivers state-of-the-art performance in general visual understanding, multimodal reasoning, and GUI grounding applications.

**VR-Thinker-7B** (Wang et al., 2025c) proposes a "thinking-with-image" framework that equips reward models with active visual reasoning operations and configurable memory windows. Through a reinforcement fine-tuning pipeline involving rejection sampling and Group Relative Policy Optimization (GRPO), it significantly improves reasoning fidelity and accuracy on video preference benchmarks, particularly for long videos.

**Video-RTS-7B** (Wang et al., 2025f) addresses the data inefficiency of RL-based video reasoning by combining pure-RL training with a video-adaptive Test-Time Scaling (TTS) strategy. Bypassing resource-intensive SFT in favor of output-based rewards and iteratively refining inference via sparse-to-dense frame addition, it surpasses existing models in accuracy while using significantly fewer training samples.

### C.3 EVALUATION METRICS

**Mean Intersection-over-Union (mIoU)** comes from Intersection-over-Union (IoU), which is a standard measure of overlap between two temporal segments. Given a predicted segment $p = [t_s^p, t_e^p]$

and a ground-truth segment $g = [t_s^g, t_e^g]$, IoU is computed as:

$$\text{IoU} = \frac{|A \cap B|}{|A \cup B|}$$

For each ground-truth segment, the maximum IoU across all predicted segments is recorded. The mean IoU (mIoU) is then obtained by averaging these values over all instances in the test set. mIoU provides a holistic measure of temporal localization accuracy, reflecting how closely predictions align with annotated spans. It is sensitive to both prediction boundary precision and temporal coverage, making it particularly suitable for localization evaluation in long-form videos.

**Recall**@$K$ assesses whether ground-truth segments are successfully retrieved by model predictions at varying strictness levels. Specifically, for a ground-truth span $g$, if there exists a prediction $p$ such that $\text{IoU}(p, g) \geq K$, the ground-truth is considered recalled. Recall@$K$ is then the fraction of recalled spans across all annotations. Typically, $K \in \{0.3, 0.5\}$ is used, where Recall@0.3 emphasizes coarse localization (lenient overlap) and Recall@0.5 emphasizes fine-grained alignment (stricter overlap). This metric complements mIoU by quantifying success rates under different quality thresholds, highlighting trade-offs between coverage and precision.

**BLEU@1 (Papineni et al., 2002)** comes from BLEU (Bilingual Evaluation Understudy), which is one of the earliest and most influential metrics for text generation evaluation. BLEU@1 focuses on unigram precision, i.e., the proportion of generated words appearing in reference captions. Formally,

$$\text{BLEU@1} = \min\left(1, \exp\left(1 - \frac{\text{len(reference)}}{\text{len(candidate)}}\right)\right) \cdot \frac{\sum_{unigram \in \text{candidate}} \text{Count}_{\text{clip}}(\text{unigram})}{\sum_{unigram \in \text{candidate}} \text{Count}(\text{unigram})}$$

The score ranges from 0 to 1, with higher scores indicating stronger lexical overlap. Although BLEU@1 provides a straightforward measure of word-level accuracy, it does not capture semantic adequacy or fluency beyond exact token matches. In video captioning, it remains useful as a proxy for surface-level similarity, particularly for frequent objects and actions.

**METEOR (Banerjee & Lavie, 2005)** (Metric for Evaluation of Translation with Explicit ORdering) addresses several limitations of BLEU by combining unigram precision and recall, alongside synonymy, stemming, and paraphrase matching. The score is computed as a harmonic mean of precision and recall (with recall typically weighted higher), and adjusted with a fragmentation penalty to account for word order:

$$\text{METEOR} = (1 - \text{Penalty}) \times F_{\text{mean}}$$

where $F_\alpha$ balances precision and recall, and $Penalty$ penalizes disordered matches. METEOR ranges from 0 to 1, yielding higher values when generated captions are both semantically complete and linguistically coherent. Its ability to match semantically related words makes it suited for evaluating paraphrased or stylistically varied captions.

**ROUGE-L (Lin, 2004)** comes from ROUGE (Recall-Oriented Understudy for Gisting Evaluation) metrics, which are widely applied in summarization and captioning. ROUGE-L specifically uses the Longest Common Subsequence (LCS) between candidate and reference sequences to compute recall, precision, and an F1-like score:

$$ROUGE - L = \frac{\sum_{S \in \{ReferenceSummaries\}} \sum_{gram_n \in S} Count_{match}(gram_n)}{\sum_{S \in \{ReferenceSummaries\}} \sum_{gram_n \in S} Count(gram_n)}$$

Here, Precision and Recall are based on the length of the LCS relative to the candidate and reference lengths, respectively. The metric rewards captions that preserve overall sentence structure and ordering of key tokens. Unlike BLEU@1, which prioritizes exact n-gram matches, ROUGE-L emphasizes global sequence-level correspondence, providing a balanced view of content fidelity.

# D  PROMPTS

## D.1  TRAINING AND EVALUATION

> **ⓘ Prompt Template for Training and Evaluation**
>
> **System Prompt:** You are an expert video analyst tasked with solving problems based on video content. When answering a question about a video, you should carefully observe and analyze important visual clues from the videos to answer. For each important segment you notice, first observe the key visual elements, then analyze their significance using the following format: specify the time range with <time>start_time-end_time</time>, describe the key visual clues with <caption>Description of key visual clues</caption>, and provide your analysis about what this means with 'Your analysis and thoughts about this segment'. Throughout your analysis, think about the question as if you were a human pondering deeply, engaging in an internal dialogue using natural thought expressions such as 'let me think', 'wait', 'Hmm', 'oh, I see', 'let's break it down', etc, or other natural language thought expressions. After examining the key visual clues, continue with deeper reasoning that connects your observations to the answer. Self-reflection or verification in your reasoning process is encouraged when necessary, though if the answer is straightforward, you may proceed directly to the conclusion. Finally, conclude by placing your final answer in <answer> </answer> tags.
> **Question Template:** {Question}
> Please analyze the video carefully by identifying key segments and their important visual clues within<time> </time>, <caption> </caption>, <think> </think> tags. Then conduct deep analysis and reasoning to arrive at your answer to the question. Finally, provide only the single option letter (e.g., A, B, C, D, E, F etc.) within the <answer> </answer> tags. Follow the format specified in the instructions.

## D.2  VIDEO CAPTION GENERATION

> **ⓘ Prompt Template for Video Caption Generation**
>
> **System Prompt:** You are a professional video analysis assistant. Your task is to analyze video segments and provide natural, factual descriptions of the key visual evidence that supports the correct answer to the given question. Focus on describing the essential visual elements, actions, objects, or events that are directly relevant to the question and answer. Provide clear, objective descriptions of what you observe without any reasoning or analysis – simply describe the important visual clues that are present in the video. Avoid referring to the content as 'this video' or adding any reasoning and thinking – instead, describe what you see directly.
> **User Prompt:** {Question} {Answer}
> Based on the video segment shown, provide a natural and concise description of the key visual evidence that supports the correct answer. Focus on describing the essential visual elements, actions, objects, or details that are directly relevant to both the question and the correct answer. Describe what you observe factually without any reasoning or analysis – simply state the important visual clues that are present. Write in a natural, descriptive style without referring to 'this video' or 'video segment'.

## D.3  QA GENERATION

> **ⓘ Prompt Template for ActivityNet QA Generation**
>
> **System Prompt:** You are an expert at creating sophisticated multiple-choice questions that test video comprehension through analysis of key visual segments.
> You will receive: 1. Background context describing the overall video content 2. A chronologically ordered list of event descriptions corresponding to key visual segments in the video
> Your task is to generate one multiple-choice question that requires viewers to locate, synthesize, and reason across these multiple key visual segments to determine the correct answer.
> Question generation strategy:
> - If events show clear relationships or logical connections: Create a reasoning question that tests understanding of cause-effect relationships, intentions, motivations, or sequential logic
> - If events appear disconnected or simple: Create a complex perceptual question that tests detailed observation, accurate pattern recognition, or comprehensive summarization across segments.

Requirements for your question: - Ask directly and naturally without referencing 'based on', 'events', 'segments', or 'sequences'
- Must require analysis of multiple event descriptions from different visual segments
- Cannot be answerable from any single event description alone
- Should demand synthesis of information across the chronological sequence
- Must test either analytical reasoning or sophisticated perceptual skills
- Base your question strictly on the information provided in the key visual segment descriptions – do not introduce any external knowledge, assumptions, or fabricated details
Requirements for answer options:
- Provide 4–6 options with one definitively correct answer
- Include sophisticated distractors that require careful discrimination
- Ensure the correct answer emerges only through comprehensive analysis of all provided events
- All options must be derivable from or directly contradicted by the given descriptions
- Avoid directly quoting phrases from the event descriptions
Output format: Respond with a valid JSON object containing these exact keys: 'question', 'options', 'answer'. The 'options' value must be a list of strings.
**User Prompt:** Background: {caption}
Descriptions of Key Visual Segments (chronological order): {events text}
Generate a multiple-choice question that requires viewers to locate and synthesize information across these specific segments.

---

### ⓘ Prompt Template for YouCook2 QA Generation

**System Prompt:** You are an expert at creating sophisticated multiple-choice questions that test cooking video comprehension through analysis of key visual segments.
You will receive: A chronologically ordered list of cooking step descriptions corresponding to key visual segments in the cooking video.
Your task is to generate one multiple-choice question that requires viewers to locate, synthesize, and reason across these multiple key visual segments to determine the correct answer.
Question generation strategy:
- You can create a reasoning question that tests understanding of cause-effect relationships, cooking techniques, ingredient interactions, or sequential cooking logic
- Or you can create a complex perceptual question that tests detailed observation, accurate pattern recognition, or comprehensive summarization across segments
Requirements for your question:
- Ask directly and naturally without referencing 'based on', 'steps', 'segments', or 'sequences'
- Must require analysis of multiple cooking step descriptions from different visual segments
- Cannot be answerable from any single step description alone
- Should demand synthesis of information across the chronological cooking sequence
- Must test either analytical reasoning or sophisticated culinary perceptual skills
- Base your question strictly on the information provided in the key visual cooking step descriptions – do not introduce any external knowledge, assumptions, or fabricated details
Requirements for answer options:
- Provide 4–6 options with one definitively correct answer
- Include sophisticated distractors that require careful discrimination
- Ensure the correct answer emerges only through comprehensive analysis of all provided cooking steps
- All options must be derivable from or directly contradicted by the given descriptions
- Avoid directly quoting phrases from the cooking step descriptions
Output format: Respond with a valid JSON object containing these exact keys: 'question', 'options', 'answer'. The 'options' value must be a list of strings.
**User Prompt:** Descriptions of Key Video Segments about Cooking Steps (chronological order): {steps text}
Generate a multiple-choice question that requires viewers to locate and synthesize information across these specific segments.

---

### ⓘ Prompt Template for TutorialVQA QA Generation

**System Prompt:** You are an expert at creating sophisticated multiple-choice questions that test video comprehension through analysis of key visual segments.
You will receive:
1. Video Title: The title of the video

2. Transcript: The spoken content or narration from the video
3. Descriptions of key video segments of main steps covered: A chronologically ordered list of step descriptions corresponding to key visual segments in the video
Your task is to generate one multiple-choice question that requires viewers to locate, synthesize, and reason across these multiple key visual segments to determine the correct answer.
Question generation strategy:
- You can create a reasoning question that tests understanding of cause-effect relationships, intentions, motivations, or sequential logic
- Or you can create a complex perceptual question that tests detailed observation, accurate pattern recognition, or comprehensive summarization across segments
Requirements for your question:
- Ask directly and naturally without referencing 'based on', 'steps', 'segments', or 'sequences'
- Must require analysis of multiple step descriptions from different visual segments
- Cannot be answerable from any single step description alone
- Should demand synthesis of information across the chronological sequence
- Must test either analytical reasoning or sophisticated perceptual skills
- Base your question strictly on the information provided in the key visual segment descriptions – do not introduce any external knowledge, assumptions, or fabricated details
Requirements for answer options: - Provide 4–6 options with one definitively correct answer
- Include sophisticated distractors that require careful discrimination
- Ensure the correct answer emerges only through comprehensive analysis of all provided steps
- All options must be derivable from or directly contradicted by the given descriptions
- Avoid directly quoting phrases from the step descriptions
Output format: Respond with a valid JSON object containing these exact keys: 'question', 'options', 'answer'. The 'options' value must be a list of strings.
**User Prompt:** Video Title: {video title}
Full Transcript: {full transcript text}
Descriptions for key video segments of main steps covered (chronological order): {main steps}
Generate a multiple-choice question that requires viewers to locate and synthesize information across these specific segments.

Table 5: Performance comparisons for Video-R1 when augmented with different chain-of-thought components: "grounding" and "captioning" CoT content. Additionally, we include ablation studies that assess the performance contributions of our synthetic captions.

| Experimental Setup | Accuracy |
|---|---|
| Base | 37% |
| w/ Grounding | 53% |
| w/ Caption | 56% |
| w/ Grounding + Caption | 63% |
| w/ Synthetic Caption | 46% |
| w/ Grounding + Synthetic Caption | 55% |

# E EXPERIMENTAL VERIFICATION OF GROUNDING AND CAPTIONING CAPABILITIES

To investigate the impact of incorporating grounding and captioning information on video reasoning performance, we conduct comprehensive experiments using Video-R1-7B (Feng et al., 2025) as our test model on the Video-Holmes (Cheng et al., 2025) dataset. This dataset provides rich annotations, including question-relevant key temporal segments (grounding information) and comprehensive video descriptions (captioning information). We evaluate the model under four distinct experimental configurations: (i) Base: Direct inference without any additional input information, serving as our baseline; (ii) w/ Grounding: Each question is augmented with temporally-grounded key segment information that highlights relevant video portions; (iii) w/ Captioning: Each question is supplemented with comprehensive caption information describing the entire video content; (iv) w/ Grounding & Captioning: Questions are enhanced with both temporal grounding and captioning information. We employ accuracy as our primary evaluation metric to assess reasoning performance across all configurations.

Table 6: Performance change of Video-Thinker with different training steps. The best results are marked in **red bold** and the second best in blue.

| Training Step | Out of Domain | | | In Domain | | | | | Avg. |
|---|---|---|---|---|---|---|---|---|---|
| | **Video-Holmes** | **CG-Bench-Reasoning** | **VRBench** | **ActivityNet** | **Star** | **ScaleLong** | **YouCook2** | **LVBench** | |
| 500 | 37.40% | 29.03% | 73.40% | 77.04% | 63.58% | 44.48% | 69.85% | 38.05% | 54.10% |
| 1000 | 38.32% | 30.30% | 71.81% | 78.16% | 68.06% | 43.53% | 69.08% | 35.35% | 54.33% |
| 1500 | 41.86% | 32.99% | 80.03% | 78.56% | 64.78% | 48.26% | **74.43%** | 37.71% | 57.33% |
| 2000 | 40.94% | 30.83% | 74.80% | **80.96%** | 62.39% | 46.06% | 68.32% | 38.38% | 55.34% |
| 2500 | **43.22%** | **33.25%** | 80.69% | 78.72% | 70.66% | **49.53%** | 73.66% | 37.04% | **58.35%** |
| 3000 | 39.36% | 32.46% | 79.33% | 78.72% | 67.16% | 48.58% | 64.12% | 36.36% | 55.76% |
| 3500 | 40.56% | 31.36% | 79.73% | 80.24% | 68.36% | 47.63% | 66.79% | 38.05% | 56.59% |
| 4000 | 41.21% | 32.84% | 79.44% | 80.00% | 70.15% | 46.69% | 66.41% | **38.72%** | 56.93% |
| 4500 | 41.92% | 32.93% | **81.79%** | 80.88% | 69.25% | 48.26% | 69.85% | 36.70% | 57.70% |
| 5000 | 41.26% | 32.01% | 78.79% | 80.72% | **71.64%** | 49.21% | 70.23% | 36.36% | 57.53% |

Table 7: Performance change of Video-Thinker with different learning rates. The best results are marked in **red bold** and the second best in blue.

| Model | LR | Out of Domain | | | In Domain | | | | |
|---|---|---|---|---|---|---|---|---|---|
| | | **Video-Holmes** | **CG-Bench-Reasoning** | **VRBench** | **ActivityNet** | **Star** | **ScaleLong** | **YouCook2** | **LVBench** |
| Qwen2.5-VL-7B-Instruct | - | 34.02% | 27.10% | 63.42% | 70.96% | 69.25% | 40.06% | 63.74% | 33.33% |
| Video-R1-7B | - | 38.54% | 27.81% | 69.25% | 76.00% | 67.76% | 47.32% | 65.65% | 34.68% |
| Video-Thinker-7B | 1e-6 | 39.14% | 28.97% | 72.79% | **80.08%** | 63.88% | 46.37% | 66.79% | 36.70% |
| | 3e-6 | 36.91% | 24.45% | 77.18% | 73.20% | 57.01% | 41.01% | 63.74% | 32.32% |
| | 5e-6 | **43.22%** | **33.25%** | **80.69%** | 78.72% | **70.66%** | **49.53%** | **73.66%** | **37.04%** |
| | 1e-5 | 16.44% | 6.86% | 18.74% | 21.20% | 23.58% | 15.14% | 1.14% | 16.16% |

To assess the quality and feasibility of our synthetic caption generation approach detailed in Section 3.1, we conduct the following ablation studies where synthetic information replaces ground-truth captions: (v) w/ Synthetic Caption: Questions are augmented with synthetically generated captioning information in place of ground-truth captions. (vi) w/ Grounding & Synthetic Caption: Questions incorporate both temporal grounding annotations and synthetic captioning information.

Note that we do not evaluate synthetic grounding generation, as our Video-Thinker framework does not synthesize temporal grounding information. All grounding annotations utilized in our experiments are derived from the original dataset annotations.

As shown in Table 5, both grounding and captioning information significantly enhance video reasoning performance. Captioning provides the largest individual improvement (37%→56%), while grounding contributes a substantial gain (37%→53%). The combination of both information types achieves the best performance at 63% accuracy, demonstrating clear synergistic effects. This suggests that grounding and captioning provide complementary benefits: grounding enables temporal focus on relevant segments, while captioning offers comprehensive contextual understanding.

Table 5 demonstrates that our synthetic captions significantly enhance the performance of the video reasoning model, yielding an improvement from 37% to 46%. While this performance gain is substantial, it does not fully match the results achieved using ground-truth captioning information. Furthermore, our synthetic captions exhibit complementary benefits when combined with ground-truth grounding information, resulting in additional performance gains from 46% to 55%. This suggests that synthetic captioning and temporal grounding provide synergistic contributions to video reasoning tasks.

# F ABLATION STUDIES

**Impact of Training Steps.** To investigate the impact of GRPO training steps on Video-Thinker's reasoning capabilities and generalization performance, we perform GRPO on Video-Thinker-SFT-7B

for varying steps from 500 to 5000 steps, saving checkpoints every 500 steps and evaluating each on both in-domain and out-of-domain benchmarks. As shown in Table 6, Video-Thinker achieves optimal performance at 2500 training steps with an average score of 58.35%, demonstrating superior results across most benchmarks. This peak performance at 2500 steps indicates an effective balance between sufficient learning and avoiding overfitting, as further training beyond this point leads to performance degradation on several benchmarks, particularly in out-of-domain scenarios, suggesting that excessive training steps may compromise the model's generalization ability while potentially overfitting to the training distribution.

**Impact of Learning Rate.** To investigate the impact of learning rate in GRPO on Video-Thinker's performance, we conduct GRPO training with four different initial learning rates (1e-6, 3e-6, 5e-6, 1e-5) and compare the results against the base model Qwen2.5-VL-7B-Instruct and the previous state-of-the-art Video-R1-7B across all in-domain and out-of-domain benchmarks. As demonstrated in Table 7, Video-Thinker achieves optimal performance with a learning rate of 5e-6, significantly outperforming both baseline models, including substantial improvements on out-of-domain tasks, while maintaining strong in-domain performance. Notably, the dramatic performance degradation at 1e-5 learning rate indicates that excessively high learning rates lead to training instability and poor convergence, while the moderate 5e-6 setting strikes an optimal balance between effective learning and stable optimization, enabling Video-Thinker to achieve superior video reasoning capabilities.

## G CASES

### G.1 CASES OF VIDEO-THINKER

In addition to the cases presented in Figure 4, we provide supplementary examples of Video-Thinker-7B's performance across diverse datasets in Figures 6, 7, 8, 9, 10, 11, 12, which demonstrate the model's capacity for iterative reasoning and error correction. This self-corrective behavior suggests that Video-Thinker transcends simple pattern matching and instead engages in a dynamic internal feedback mechanism.

### G.2 COMPARISONS BETWEEN VIDEO-THINKER WITH VIDEO REASONING METHODS WITH TOOL USE

As introduced in Section 4.4, a central claim of our work is that our CoT design and hindsight-curated data synthesis enhance the captioning and grounding capabilities inherent in our method. To validate this claim, we conduct a comparative analysis examining whether external tool integration can achieve comparable performance improvements.

We compare Video-Thinker against several baseline configurations that augment the base model Qwen2.5-VL-7B-Instruct with external specialized tools: the grounding model Temporal-R1-7B (Li et al., 2025b) and the captioning model Skycaptioner-V1-8B (Chen et al., 2025a). Counterintuitively, our results in Table 4 demonstrate that incorporating these external tools actually degrades the base model's performance. We hypothesize that this degradation stems from compatibility issues between the specialized models and Qwen2.5-VL-7B-Instruct. To test this hypothesis, we construct an alternative baseline where Qwen2.5-VL-7B and its larger counterpart Qwen2.5-VL-72B serve as both grounding and captioning modules. However, as shown in Table 4, this configuration also fails to improve upon the base model's capabilities, indicating that the issue extends beyond model compatibility.

To understand this counterintuitive finding, we conduct case studies on the Video-Holmes dataset. Through the analysis presented in Figure 15 (Appendix G.2), we identify the underlying mechanisms of failure. Specifically, we observe that Qwen2.5-VL-7B-Instruct with 72B as both grounding and captioning tools misinterpret the caption describing "the scene darkening" as a cinematographic technique rather than recognizing it as describing an actual event (lights turning off). This misinterpretation leads the model to select "hallucination" as an irrelevant answer instead of identifying the physical cause of the lighting change.

These case studies reveal that CoT information from external tools can mislead the base model when the reasoning chain becomes discontinuous. The systematic risks in tool invocation and result

**Video:**

**Question:** What is the correct order of the following events?\n①Netizens send secretly taken photos\n②Kira starts live interaction\n③The door automatically opens\n④Receive threatening comments with an address\n⑤Dial the police
  A. ②④①⑤③  B. ④②①③⑤  C. ②①④⑤③  D. ④⑤②①③  E. ②④⑤①③  F. ①④②⑤③

**Qwen2.5-VL-7B:** Let's break down the events step by step based on the visual clues and the sequence of events:
1. ② Kira starts live interaction: This is the initial event where Kira begins her live interaction, as indicated by the "LIVE" indicator and the chat messages.
2. ④ Receive threatening comments with an address: After Kira starts her live interaction, she receives threatening comments, including an address, as shown in the chat messages.
3. ① Netizens send secretly taken photos: Following the threatening comments, netizens send secretly taken photos, as indicated by the visual of the photo frame.
4. ⑤ Dial the police: After receiving the threatening comments and photos, Kira decides to dial the police for help.
5. ③ The door automatically opens: Finally, the door automatically opens, possibly indicating that the police or someone else has arrived to help.
Given this sequence, the correct order of events is ②④①⑤③.
<answer>C</answer> ❌

**Video-R1-7B:** <think>Let me think about this carefully. The question is asking for the correct order of events based on the video content. The events are numbered and we need to match them with the correct sequence.
First, let's break down the events:
1. Netizens send secretly taken photos
2. Kira starts live interaction
3. The door automatically opens
4. Receive threatening comments with an address
5. Dial the police
Now, let's analyze the video content:- Kira starts live interaction first, which is event ②.- After the live interaction, netizens send secretly taken photos, which is event ①.- Then, Kira receives threatening comments with an address, which is event ④.- After that, Kira dials the police, which is event ⑤.- Finally, the door automatically opens, which is event ③.
So, the correct order should be ②①④⑤③.
Let's verify this with the options:A. ②④①⑤③ - This is not correct because ④ is before ①.B. ④②①③⑤ - This is not correct because ② is before ④.C. ②①④⑤③ - This matches our analysis.D. ④⑤②①③ - This is not correct because ② is before ④.E. ②④⑤①③ - This is not correct because ① is before ④.F. ①④②⑤③ - This is not correct because ② is before ①.The correct order is ②①④⑤③, which corresponds to option C.</think>
<answer>C</answer> ❌

**Video-Thinker-7B:** Let me watch this video carefully and piece together the sequence of events.
<time>00:00-00:15</time>
<caption>A woman in a Santa hat is speaking into a microphone. The words "KIRA" appear on the screen, and the number "43" is visible in the corner. The background features a wooden wall with colorful lights.</caption>
<think>Okay, so this is the start of the video - Kira is clearly the main subject here. The setup looks like a live interaction or stream, given the microphone and the "LIVE" indicator.</think>Hmm, let's see what happens next...
<time>00:15-00:30</time>
<caption>The woman continues speaking, and the words "who will pay to take kira home for christmas?" appear on the screen. The number "44" is now visible.</caption>
<think>Ah, this seems like an interactive question or prompt - it's part of the live interaction. The number changing suggests this is a live feed with real-time engagement.</think>Wait, there's more context coming up...
<time>00:30-00:45</time>
<caption>The woman is still speaking, and the words "user1234" appear on the screen. The number "44" remains visible.</caption>
<think>Interesting - this seems to be a comment or interaction from a viewer. The "user1234" tag confirms this is part of the live chat or comments section.</think>Now let's see how the situation develops...
<time>00:45-01:00</time>
<caption>The woman continues speaking, and the words "i know where you live" appear on the screen. The number "44" is still visible.</caption>
<think>Oh no, this is a threatening comment! The woman is clearly reacting to this dangerous message. The live interaction is escalating.</think>Wait, there's a crucial moment coming up...
<time>01:00-01:15</time>
<caption>The woman is still speaking, and the words "dial the police" appear on the screen. The number "44" remains visible.</caption>
<think>Ah-ha! This is the key moment - she's responding to the threat by calling for help. The live interaction has reached a dangerous point.</think>Finally, let's see how it concludes...
<time>01:15-01:30</time>
<caption>The woman continues speaking, and the words "the door automatically opens" appear on the screen. The number "44" is still visible.</caption><think>Wow, this is unexpected! The door opening seems like a sudden development, possibly related to the threat. The live interaction is still ongoing.</think>Now let's match this sequence to the options:
1. Kira starts live interaction (matches the first segment)
2. Receive threatening comments with an address (matches the second segment)
3. Netizens send secretly taken photos (matches the third segment)
4. Dial the police (matches the fourth segment)
5. The door automatically opens (matches the fifth segment)
Looking at the options:- A matches perfectly: ②④①⑤③- B has the door opening too early- C has the door opening before dialing the police- D starts with the door opening, which is wrong- E starts with netizens sending photos, which isn't first- F starts with netizens photos, which is wrongThe answer is clearly A. The sequence follows exactly what we saw in the video: live interaction first, then threatening comments, then photos being sent, then calling the police, and finally the door opening.
<answer>A</answer> ✅

Figure 6: An example of Video-Thinker-7B's reasoning output on Video-Holmes dataset

integration suggest that effective tool utilization requires sophisticated coordination mechanisms to maintain reasoning coherence across modular components.

**Video:**

**Question:** What is the correct order of the following events?\n①The vehicle broke down. ②The red-haired dog looked at the photo frame. ③The blond man was possessed and pushed people. ④The skeleton disappeared. ⑤Enter the castle.
A. ①⑤③②④  B. ①⑤②③④  C. ⑤①②③④  D. ⑤①③②④  E. ①③⑤②④  F. ⑤②①③④

**Qwen2.5-VL-7B:** Let's break down the events step by step based on the video content:
1. Enter the castle: The video starts with the characters entering a castle, which is a key event.
2. The vehicle broke down: After entering the castle, the vehicle breaks down, which is the next event.
3. The red-haired dog looked at the photo frame: The dog looks at a photo frame, which is a subsequent event.
4. The blond man was possessed and pushed people: The blond man is possessed and pushes people, which is another event.
5. The skeleton disappeared: Finally, the skeleton disappears, which is the last event.
Now, let's match these events with the given options:- Option A: ①⑤③②④- Option B: ①⑤②③④- Option C: ⑤①②③④- Option D: ⑤①③②④- Option E: ①③⑤②④- Option F: ⑤②①③④The correct order is ⑤①②③④.
<answer>C</answer> ❌

**Video-R1-7B:** <think>Let me think about this carefully. The question is asking for the correct order of events based on the video content. I need to identify the key segments and their order in the video.
First, let's break down the events:
1. The vehicle broke down.2. The red-haired dog looked at the photo frame.3. The blond man was possessed and pushed people.4. The skeleton disappeared.5. Enter the castle.
Now, let's analyze the video:- The vehicle breaking down is the first event. This is clearly shown in the video.- After the vehicle breaks down, the characters enter the castle. This is the next logical step.- The red-haired dog looking at the photo frame happens after entering the castle.- The blond man being possessed and pushing people happens after the dog looks at the photo frame.- The skeleton disappearing is the last event.So, the correct order should be:1. The vehicle broke down.2. Enter the castle.3. The red-haired dog looked at the photo frame.4. The blond man was possessed and pushed people.5. The skeleton disappeared.
This corresponds to option C.
Let's verify:
- The vehicle breaking down is first.- Entering the castle is second.- The dog looking at the photo frame is third.- The man being possessed is fourth.- The skeleton disappearing is fifth.
Everything seems to fit the order in option C.
Oh, I see. The correct answer is C.</think>
<answer>C</answer> ❌

**Video-Thinker-7B:** Let me watch this video carefully and piece together the sequence of events.
<time>00:00-00:10</time>
<caption>A red-haired dog looks at a photo frame on a wall.</caption>
<think>Okay, so the first thing we see is the red-haired dog examining a photo frame. That must be the first event.</think>
Hmm, what happens next?
<time>00:10-00:20</time>
<caption>A vehicle with a blue character drives through a forest.</caption>
<think>Wait, this seems like the second event - the vehicle is moving through the forest. The blue character is likely the driver.</think>
Now let's see what comes after that...
<time>00:20-00:30</time>
<caption>The vehicle breaks down and the blue character gets out.</caption>
<think>Ah, here we see the vehicle breaking down. That's the third event.</think>
Interesting, what happens next?
<time>00:30-00:40</time>
<caption>A blond man in a suit is possessed and pushes people.</caption>
<think>Now we see the possessed man - this must be the fourth event.</think>
Finally, what's the last thing we see?
<time>00:40-00:50</time>
<caption>A skeleton disappears from a frame on the wall.</caption>
<think>And the last event is the skeleton disappearing from the photo frame.</think>Now let's match this sequence to the options:1. The red-haired dog looking at the photo frame (first event)2. The vehicle breaking down (second event)3. The possessed man pushing people (third event)4. The skeleton disappearing (fourth event)5. Entering the castle (last event)
Looking at the options:
- A has the vehicle breaking down first, which matches- B has the vehicle breaking down second, which is wrong- C starts with entering the castle, which is incorrect- D starts with entering the castle, which is wrong- E starts with the vehicle breaking down, but then skips the first event- F starts with entering the castle, which is wrongThe correct sequence is B: ①⑤②③④. The other options either start with the wrong event or rearrange the sequence incorrectly.The answer is B.
<answer>B</answer> ✅

Figure 7: An example of Video-Thinker-7B's reasoning output on Video-Holmes dataset

# H LIMITATIONS

Despite improvements in temporal reasoning, Video-Thinker's performance on extremely long videos (>1 hour) remains constrained by the inherent context limitations of the underlying language model architecture. The model's ability to maintain coherent reasoning across very extended temporal sequences may degrade, particularly when critical information is distributed across distant temporal segments. Also, the generation of explicit grounding, captioning, and reasoning tokens results in longer inference sequences. This inevitably leads to higher latency and computational costs compared to models that output direct answers without intermediate steps. Moreover, the current framework focuses exclusively on visual and textual modalities, lacking integration of audio information that could provide crucial contextual cues for comprehensive video understanding. This limitation may

be particularly pronounced in scenarios where audio-visual synchronization or speech content is essential for accurate reasoning.

## I    USE OF LLMS

During the preparation of this manuscript, we made limited use of publicly available large language models (LLMs) to assist with English writing. All technical content, including the formulation of ideas, design of methodologies, implementation of experiments, and interpretation of results, was entirely conceived and written by the authors without the involvement of LLMs. The role of LLMs was strictly confined to stylistic and linguistic improvements, in a manner comparable to grammar- or spell-checking software. We ensured that no novel research insights, data, or analyses were generated by LLMs, and all scientific claims and results presented in this work remain the sole responsibility of the authors.

**Video:**

Question: Why did Team B4NTI choose to make a tactical change in the game?
    A. Because Team B4NTI noticed that Team INVIL was vulnerable to a high-pressing strategy.
    B. Because Team B4NTI wanted to experiment with a new formation they had developed in training.
    C. Because Team B4NTI realized that their previous tactics were not enough to beat Team INVIL.
    D. Because Team B4NTI needed to adapt to the unexpected weather conditions affecting the pitch.

**Qwen2.5-VL-7B:** Let's break down the analysis step by step:
1. Initial Situation: The video shows a match between Team B4NTI and Team INVIL. The score is tied at 0-0, and the game is in the early stages.
2. Visual Clues: The video shows a map with various locations marked, and the players are discussing their strategy. There is a mention of a "tactical change" being made by Team B4NTI.
3. Key Visual Clues:
   - The map shows different areas, and the players are discussing which areas to focus on.
   - The players are seen looking at the map and discussing their strategy, indicating a strategic decision is being made.
   - The players mention "tactical change" and "high-pressing strategy," suggesting they are adapting their approach to the game.
4. Reasoning:
   - The players are not experimenting with a new formation (option B) because they are discussing a specific strategy rather than a new one.
   - The weather conditions are not mentioned, so option D can be ruled out.
   - The players are not just realizing that their previous tactics were not enough (option C) because they are actively discussing and making a change.
   - The players are discussing a "high-pressing strategy," which implies they are noticing a vulnerability in Team INVIL's defense and are planning to exploit it.
5. Conclusion:
   - The most logical reason for the tactical change is that Team B4NTI noticed that Team INVIL was vulnerable to a high-pressing strategy.
Therefore, the answer is:
<answer>A</answer> ❌

**Video-R1-7B:** <think>Let's break down the situation and the options provided. The video shows a match between Team B4NTI and Team INVIL, and Team B4NTI made a tactical change during the game. The question is asking why they chose to make this change.
First, let's consider the options:
A. Because Team B4NTI noticed that Team INVIL was vulnerable to a high-pressing strategy.
B. Because Team B4NTI wanted to experiment with a new formation they had developed in training.
C. Because Team B4NTI realized that their previous tactics were not enough to beat Team INVIL.
D. Because Team B4NTI needed to adapt to the unexpected weather conditions affecting the game.
From the video, we can see that Team B4NTI was already in a competitive position, and they made a tactical change. This suggests that they were not just experimenting or reacting to weather conditions. Instead, they likely noticed something specific about Team INVIL's defense or their own strengths.
Option A seems the most plausible. The video shows that Team B4NTI was able to capitalize on a weakness in Team INVIL's defense, which led to a tactical change. This aligns with the observation that they were already in a competitive position and needed to exploit a specific vulnerability.
Option B is less likely because there is no indication that they were experimenting with a new formation. They were already in a competitive position and needed a specific change.
Option C is also less likely because they were already in a competitive position and needed a specific change, not just realizing their previous tactics were not enough.
Option D is not relevant to the tactical change in the game.
Therefore, the most logical reason for Team B4NTI to make a tactical change is that they noticed a vulnerability in Team INVIL's defense.
</think>
<answer>A</answer> ❌

**Video-Thinker-7B:** Let's break this down step by step to understand why Team B4NTI made that tactical change.
<time>00:00-00:10</time>
<caption>The scoreboard shows Team INVIL leading with a score of 2-0.</caption>
<think>Okay, so Team INVIL is already in control with a 2-0 lead. This suggests Team B4NTI is behind and needs to make a change to catch up.</think>
Hmm, let's see what happens next...
<time>00:10-00:20</time>
<caption>The players from Team B4NTI are seen discussing strategy among themselves.</caption>
<think>This is interesting - they're clearly analyzing the situation and deciding on a new approach. The fact that they're talking about strategy suggests they're making a deliberate change rather than just reacting to something.</think>
Wait, there's more context here...
<time>00:20-00:30</time>
<caption>The scoreboard updates to show Team B4NTI now leading with a score of 2-1.</caption>
<think>Ah! This is the key moment. After making their change, Team B4NTI successfully reversed the lead. This shows their tactical adjustment was effective.</think>
Now let's evaluate the options:
A) "Because Team B4NTI noticed that Team INVIL was vulnerable to a high-pressing strategy" - There's no evidence of INVIL being vulnerable or B4NTI using a high-pressing strategy. The change was more about reversing the lead.B) "Because Team B4NTI wanted to experiment with a new formation they had developed in training" - While they might have been experimenting, the change was clearly tied to the game situation, not just training.C) "Because Team B4NTI realized that their previous tactics were not enough to beat Team INVIL" - This matches perfectly. They saw they were behind (2-0) and needed a change to win (2-1).D) "Because Team B4NTI needed to adapt to the unexpected weather conditions affecting the pitch" - There's no mention of weather in the video segments.
The answer is clearly C - they recognized their previous tactics weren't working and needed to make a strategic change to win. The other options either don't match or are too vague given the specific game situation shown.
The answer is C.
<answer>C</answer> ✅

Figure 8: An example of Video-Thinker-7B's reasoning output on VRBench dataset

**Video:**

**Question:** Multiple scenes in the video show people gathering around the campsite flames, why are they doing this?
A. They may be doing preparatory work for subsequent games, waiting for the interface, for skills and weapons to be selected.
B. They are participating in a ceremony to unlock hidden abilities and special items.
C. They are engaging in a ritual to enhance their character attributes before the next challenge.
D. They are coordinating strategies and discussing team formations for upcoming battles.

**Qwen2.5-VL-7B:** Let's break down the analysis step by step:
1. Identify the Setting: The video shows a campsite with a fire, which is a common setting in many video games, especially those involving fantasy or survival elements. The presence of multiple characters gathered around the fire suggests a social or communal activity.
2. Visual Clues: The characters are standing close to each other, which is typical in video games for social interactions. The campfire is a central element, often used in games to signify a gathering point or a moment of rest and socialization.
3. Behavior of Characters: The characters appear to be engaged in a discussion or interaction, which is a common behavior in video games when players gather around a campfire. This could be for strategizing, socializing, or preparing for the next phase of the game.
4. Contextual Clues: The video does not show any specific actions that would indicate a ritual, ceremony, or preparation for a game interface. The focus seems to be on the characters interacting with each other rather than engaging in a specific activity.
5. Elimination of Options:- Option A: This option talks about preparatory work for games, which is not supported by the visual clues. The characters are not shown interacting with game interfaces or preparing for a game.   - Option B: This option suggests a ceremony to unlock hidden abilities. There is no indication of any special items or abilities being unlocked in the video.   - Option C: This option mentions a ritual to enhance character attributes. There is no evidence of any ritualistic actions or enhancements being performed.- Option D: This option suggests that the characters are coordinating strategies and discussing team formations. This is a plausible scenario given the social nature of the gathering around the campfire.
Based on the analysis, the most fitting option is:
\<answer\>D\</answer\> ❌

**Video-R1-7B:** \<think\>Let's break down the video scenes and the context provided. The video shows a group of people gathered around a campsite, which is a common setting for various activities in video games. The presence of a campfire and the casual gathering suggests a preparatory phase rather than an immediate action.
First, let's consider the options:
A. They may be doing preparatory work for subsequent games, waiting for the interface, for skills and weapons to be selected.B. They are participating in a ceremony to unlock hidden abilities and special items.C. They are engaging in a ritual to enhance their character attributes before the next challenge.D. They are coordinating strategies and discussing team formations for upcoming battles.
From the video, it's clear that the group is not actively engaged in combat or a specific ritual. They are standing around the campfire, which is a typical setting for players to gather and strategize. The campfire provides a communal space where players can discuss their plans and coordinate their actions.
The group's posture and the fact that they are standing around the campfire suggest that they are likely discussing their next moves and strategies. This aligns with option D, which focuses on coordination and strategy.
Now, let's verify if the other options fit the context:- Option A is possible, but it doesn't fully capture the communal aspect of the gathering.- Option B is less likely because there is no indication of a ceremony or unlocking abilities.- Option C is also less likely because there is no ritualistic behavior observed.After this analysis, it's clear that the most fitting option is D, as it aligns with the observed behavior and the typical behavior of players in video games.
Therefore, the answer is D.\</think\>
\<answer\>D\</answer\> ❌

**Video-Thinker-7B:** Let's break this down step by step to understand what's happening in the video.
\<time\>00:00-00:15\</time\>
\<caption\>A group of people are gathered around a campsite with flames burning brightly. They are standing close together, some holding torches and others with weapons. The scene is illuminated by the firelight, casting shadows on their faces.\</caption\>
\<think\>Okay, so we have a group of people at what looks like a campsite, with flames and torches. The atmosphere seems focused and purposeful - not just casual hanging out. The weapons and torches suggest some kind of preparation or ritual.\</think\>
Hmm, let's see what happens next...
\<time\>00:15-00:30\</time\>
\<caption\>The people are now gathered around a campfire, still holding their weapons and torches. They appear to be discussing something, with some gesturing towards the fire. The scene is still illuminated by the firelight, creating a dramatic effect.\</caption\>
\<think\>Now they're gathered around the campfire, still holding weapons and torches. The discussion seems important - they're gesturing and looking at each other seriously. The firelight is creating a dramatic, almost ceremonial atmosphere. This doesn't seem like casual strategy discussion (option D), and the weapons and torches don't seem like they're for battle (option B).\</think\>
Wait, there's more context...
\<time\>00:30-00:45\</time\>
\<caption\>The scene shifts to show a character interface with various options and statistics. The group is still gathered around the campfire, now with some holding up their phones to view the interface. The character interface appears to be part of a game system.\</caption\>
\<think\>Ah! This is key. The character interface with options and statistics suggests this is part of a game system. The group is clearly interacting with this interface, which seems to be part of some kind of preparation or ritual before gameplay. The weapons and torches might be part of this game system too.\</think\>
Now let's evaluate the options:
- A mentions preparatory work for games, which matches what we're seeing with the character interface and weapons/torches.- B suggests a ceremony to unlock abilities, but the focus seems more on the game system than unlocking abilities.
- C mentions enhancing character attributes, but the emphasis seems more on the game system than attribute enhancement.
- D is about strategy discussion, but the weapons and torches don't seem like battle strategy tools.
The most fitting explanation is that they're doing preparatory work for subsequent games, waiting for the interface, for skills and weapons to be selected. The character interface and weapons/torches all point to this being part of a game system.
The answer is A.
\<answer\>A\</answer\> ✅

Figure 9: An example of Video-Thinker-7B's reasoning output on VRBench dataset

Video:

Question: Why does the protagonist of the video feel comfortable with this way of getting on the boat?
- A. Because he doesn't have to swim
- B. Because the weather is nice
- C. Because the boat is moving slowly
- D. Because he can sit directly on the boat
- E. Because he is used to this method
- F. Because the boat is stable
- G. Because the boat has a comfortable seat
- H. Because he enjoys the view from the boat

**Qwen2.5-VL-7B:** Let's break down the video to understand the context and the protagonist's comfort level with the method of getting on the boat.
1. Initial Scene: The protagonist is standing on a rocky terrain, holding a large, yellow, and black object. This suggests he is preparing for an outdoor activity, possibly related to exploration or adventure.
2. Transition to Boat Scene: The video cuts to a scene where the protagonist is on a boat. He is seen sitting on the edge of the boat, which is moving through water. The water is choppy, indicating that the boat is not moving slowly or steadily.
3. Observation of the Boat: The protagonist is not sitting on a comfortable seat or a stable platform. The boat is moving, and the water is not calm, which suggests that the boat is not stable.
4. Conclusion: Given the choppy water and the lack of a stable platform, it is unlikely that the protagonist feels comfortable with this method of getting on the boat. The options provided do not align with the visual clues in the video.
Therefore, the correct answer is:
<answer>E</answer> ❌

**Video-R1-7B:** <think>Let me think about this carefully. The question is asking why the protagonist feels comfortable with a specific method of getting on the boat. The options provided give various reasons, and I need to analyze the video to determine which one fits best.
First, let's break down the options:
A. Because he doesn't have to swim - This could be a reason, but it's not explicitly mentioned in the video.B. Because the weather is nice - The weather is not a factor in the video, so this can be ruled out.C. Because the boat is moving slowly - The boat's speed is not a significant factor in the video.D. Because he can sit directly on the boat - This could be a reason, but it's not the only factor.E. Because he is used to this method - This seems plausible, as the protagonist appears comfortable and familiar with the method.F. Because the boat is stable - The stability of the boat is not a factor in the video.G. Because the boat has a comfortable seat - The comfort of the seat is not a factor in the video.H. Because he enjoys the view from the boat - The view is not a significant factor in the video.
From the analysis, the most plausible reason is that the protagonist is used to this method. He appears comfortable and familiar with the method, which suggests that he is used to it.
Hmm, after this deep analysis, it seems that the most logical reason is that the protagonist is used to this method. Therefore, the correct answer is E.
Oh, I see. The correct answer is E.</think>
<answer>E</answer> ❌

**Video-Thinker-7B:** Let me watch this video carefully to understand why the protagonist feels comfortable with this method of getting on the boat.
<time>00:00-00:10</time>
<caption>A man in a helmet and backpack stands near a large rock formation, holding a rope. He then climbs up the rope and onto a boat.</caption>
<think>Okay, so the protagonist is clearly using a rope to climb onto the boat. This seems like an unconventional or specialized way to get on board. The fact that he's comfortable doing this suggests it's something he's used to.</think>
Hmm, let's see what happens next...
<time>00:10-00:25</time>
<caption>The man continues climbing up the rope and onto the boat, eventually sitting down comfortably.</caption>
<think>Now he's clearly sitting directly on the boat after climbing up. The way he moves seems confident and practiced - like he's done this many times before. The fact that he's comfortable sitting directly on the boat is a strong clue.</think>
Wait, there's more context here...
<time>00:25-00:35</time>
<caption>The boat is moving slowly through the water, and the man remains seated comfortably.</caption>
<think>The boat's movement seems slow and steady, but this doesn't seem to bother the man at all. He's still seated comfortably, which suggests he's used to this kind of movement.</think>
Now let's evaluate the options:
A) "Because he doesn't have to swim" - There's no mention of swimming in the video, so this doesn't seem relevant.B) "Because the weather is nice" - The weather isn't shown or mentioned, so this can't be correct.C) "Because the boat is moving slowly" - The boat is moving slowly, but this doesn't explain why he's comfortable - it's just the boat's motion.D) "Because he can sit directly on the boat" - This matches perfectly with what we see - he's comfortably seated after climbing up.E) "Because he is used to this method" - This could be true, but the video doesn't explicitly show his past experience.F) "Because the boat is stable" - The boat is moving slowly, but stability isn't shown as the reason.G) "Because the boat has a comfortable seat" - The seat isn't shown or mentioned.H) "Because he enjoys the view"
- No view is shown or mentioned.The most direct evidence is that he's comfortably seated after climbing up - that's exactly what the video shows. The other options either aren't shown or aren't as directly supported.
The answer is D.
<answer>D</answer> ✅

Figure 10: An example of Video-Thinker-7B's reasoning output on CG-Bench dataset

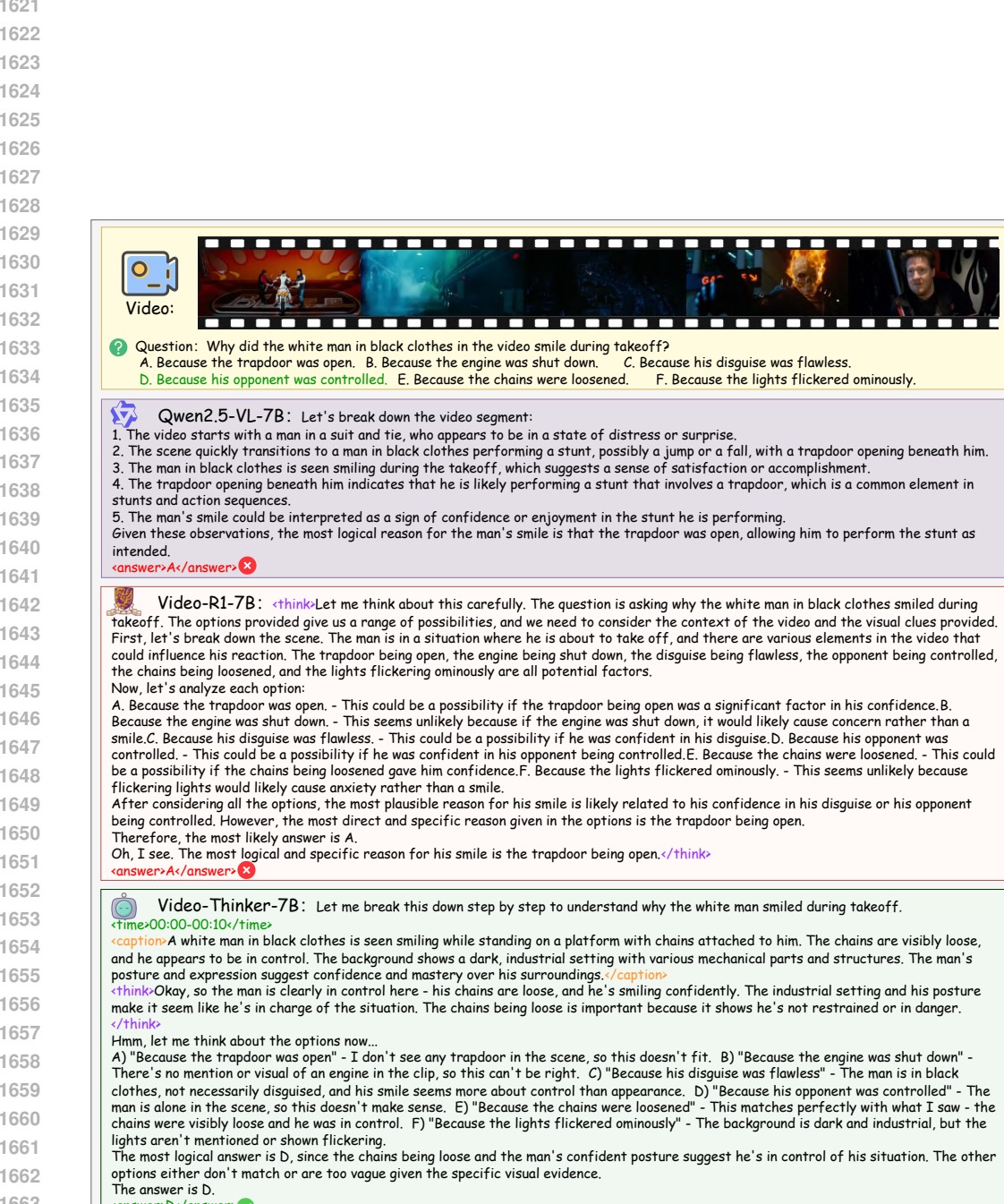

Figure 11: An example of Video-Thinker-7B's reasoning output on CG-Bench dataset

Video:

Question: What did the protagonist in the video eat for dinner at the restaurant near the hotel?
A. Tempura, salmon fillet, tofu salad    B. Miso soup, tuna roll, vegetable skewers    C. Sashimi, fried fish, chicken thigh side dishes
D. Sushi, grilled shrimp, beef teriyaki    E. Ramen, crispy squid, duck breast    F. Udon, grilled eel, pork belly bites

Qwen2.5-VL-7B: Let's analyze the video step by step:
1. The video starts with a scene of a group of cats gathered on a rooftop. This is not related to the dinner at the restaurant.2. The next scene shows a staircase leading up to a building, which could be the hotel mentioned in the question.3. The video then transitions to a restaurant interior, where a person is seen entering through a door. This suggests that the person is heading to the restaurant.4. The video then shows a close-up of a plate of food, which includes tempura, salmon fillet, and tofu salad. This is a clear indication of the meal being served.5. The video then shows a person eating the food, confirming that this is the dinner being referred to in the question.
Given the analysis, the meal at the restaurant appears to be tempura, salmon fillet, and tofu salad.
<answer>A</answer> ✗

Video-R1-7B: <think>Let me think about this carefully. The question is asking about the dinner eaten by the protagonist at a restaurant near the hotel. The options provided are quite detailed, and each one lists different types of dishes.
First, I need to recall the details of the video. The protagonist went to a restaurant near the hotel, and the video showed the dishes being served. The dishes included a variety of seafood and some side dishes.
Now, let's analyze the options:
A. Tempura, salmon fillet, tofu salad - This option includes a variety of seafood and a side dish, which seems plausible.B. Miso soup, tuna roll, vegetable skewers - This option includes a soup and some skewers, which might be part of the meal, but it's not as comprehensive as option A.C. Sashimi, fried fish, chicken thigh side dishes - This option includes sashimi, which is a type of raw fish, and some chicken, which is not as diverse as option A.D. Sushi, grilled shrimp, beef teriyaki - This option includes sushi, which is a type of rolled rice, and some grilled shrimp, but it lacks the variety of option A.E. Ramen, crispy squid, duck breast - This option includes ramen, which is a noodle dish, and some duck, which is not as diverse as option A.F. Udon, grilled eel, pork belly bites - This option includes udon, which is a noodle dish, and some grilled eel, which is not as diverse as option A.
After considering all the options, it seems that option A provides the most comprehensive and diverse set of dishes that match the details in the video.
Oh, I see. The correct answer is A, as it includes a variety of seafood and a side dish, which matches the details in the video.</think>
<answer>A</answer> ✗

Video-Thinker-7B: Let me watch this video carefully to figure out what the protagonist ate for dinner.
<time>00:00-00:15</time>
<caption>A group of cats are gathered around a person holding a plate of food.</caption>
<think>Okay, this seems like the setup - the protagonist is about to eat, and there are cats involved, which might be part of the dining experience.</think>
Hmm, let's see what's on the plate...
<time>00:15-00:30</time>
<caption>The person then eats from the plate, taking bites of what appears to be sushi.</caption>
<think>Ah! The first food item is clearly sushi. That's important - it's one of the options.</think>
Wait, there's more to this...
<time>00:30-00:45</time>
<caption>The person continues eating, now taking bites of grilled shrimp.</caption>
<think>Now we're seeing grilled shrimp being eaten. That matches one of the options too.</think>Let me check the last part...
<time>00:45-00:55</time>
<caption>The person finishes eating and the scene transitions to them walking away from the restaurant.</caption>
<think>So the entire meal was just these two items - sushi and grilled shrimp.</think>
Now looking at the options:
- A has tempura and tofu salad, which weren't shown- B has miso soup and tuna roll, neither of which appeared- C has sashimi and fried fish, but those weren't on the plate- D has sushi and grilled shrimp - this matches exactly what we saw!- E has ramen and crispy squid, neither of which were part of the meal- F has udon and grilled eel, which weren't shown
The answer is clearly C, since the video showed exactly those two items being eaten. The other options either didn't appear or were completely different from what was shown.
The answer is C.
<answer>C</answer> ✓

Figure 12: An example of Video-Thinker-7B's reasoning output on CG-Bench dataset

**Video:** GHOST

**Question:** What is the correct order of the following events?\n①The vehicle broke down. ②The red-haired dog looked at the photo frame. ③The blond man was possessed and pushed people. ④The skeleton disappeared. ⑤Enter the castle.
A. ①⑤③②④  B. ①⑤②③④  C. ⑤①②③④  D. ⑤①③②④E. ①③⑤②④    F. ⑤②①③④

**Qwen2.5-VL-7B:** Let's break down the events step by step based on the video content:
1. Enter the castle: The video starts with the characters entering a castle, which is a key event.2. The vehicle broke down: After entering the castle, the vehicle breaks down, which is the next event.3. The red-haired dog looked at the photo frame: The dog looks at a photo frame, which is a subsequent event.4. The blond man was possessed and pushed people: The blond man is possessed and pushes people, which is another event.5. The skeleton disappeared: Finally, the skeleton disappears, which is the last event.
Now, let's match these events with the given options:
- Option A: ①⑤③②④- Option B: ①⑤②③④- Option C: ⑤①②③④- Option D: ⑤①③②④- Option E: ①③⑤②④- Option F: ⑤②①③④
The correct order is ⑤①②③④.
<answer>C</answer> ❌

**Video-R1-7B:** <think>Let me think about this carefully. The question is asking for the correct order of events based on the video content. I need to identify the key segments and their order in the video.
First, let's break down the events:1. The vehicle broke down.2. The red-haired dog looked at the photo frame.3. The blond man was possessed and pushed people.4. The skeleton disappeared.5. Enter the castle.
Now, let's analyze the video:- The vehicle breaking down is the first event. This is clearly shown in the video.- After the vehicle breaks down, the characters enter the castle. This is the next logical step.- The red-haired dog looking at the photo frame happens after entering the castle.- The blond man being possessed and pushing people happens after the dog looks at the photo frame.- The skeleton disappearing is the last event.
So, the correct order should be:1. The vehicle broke down.2. Enter the castle.3. The red-haired dog looked at the photo frame.4. The blond man was possessed and pushed people.5. The skeleton disappeared.
This corresponds to option C.
Let's verify:- The vehicle breaking down is first.- Entering the castle is second.- The dog looking at the photo frame is third.- The man being possessed is fourth.- The skeleton disappearing is fifth.
Everything seems to fit the order in option C.
Oh, I see. The correct answer is C.</think>
<answer>C</answer> ❌

**Video-Thinker-7B:** Let me watch this video carefully and piece together the sequence of events.
<time>00:00-00:10</time>
<caption>A red-haired dog looks at a photo frame on a wall.</caption>
<think>Okay, so the first thing we see is the red-haired dog examining a photo frame. That must be the first event.</think>
Hmm, what happens next?
<time>00:10-00:20</time>
<caption>A vehicle with a blue character drives through a forest.</caption>
<think>Wait, this seems like the second event - the vehicle is moving through the forest. The blue character is likely the driver.</think>
Now let's see what comes after that...
<time>00:20-00:30</time>
<caption>The vehicle breaks down and the blue character gets out.</caption>
<think>Ah, here we see the vehicle breaking down. That's the third event.</think>Interesting, what happens next?
<time>00:30-00:40</time>
<caption>A blond man in a suit is possessed and pushes people.</caption>
<think>Now we see the possessed man - this must be the fourth event.</think>
Finally, what's the last thing we see?
<time>00:40-00:50</time>
<caption>A skeleton disappears from a frame on the wall.</caption>
<think>And the last event is the skeleton disappearing from the photo frame.</think>
Now let's match this sequence to the options:
1. The red-haired dog looking at the photo frame (first event)2. The vehicle breaking down (second event)3. The possessed man pushing people (third event)4. The skeleton disappearing (fourth event)5. Entering the castle (last event)
Looking at the options:- A has the vehicle breaking down first, which matches- B has the vehicle breaking down second, which is wrong- C starts with entering the castle, which is incorrect- D starts with entering the castle, which is wrong- E starts with the vehicle breaking down, but then skips the first event- F starts with entering the castle, which is wrong
The correct sequence is B: ①⑤②③④. The other options either start with the wrong event or rearrange the sequence incorrectly.The answer is B.
<answer>B</answer> ✅

Figure 13: An example of Video-Thinker-7B's reasoning output on Video-Holmes dataset

**Video**

You are an expert video analyst tasked with solving problems based on video content. When answering a question about a video, you should carefully observe and analyze important visual clues from the videos to answer. For each important segment you notice, first observe the key visual elements, then analyze their significance using the following format: specify the time range with <time>start_time-end_time</time>, describe the key visual clues with <caption>Description of key visual clues</caption>, and provide your analysis about what this means with 'Your analysis and thoughts about this segment'. Throughout your analysis, think about the question as if you were a human pondering deeply, engaging in an internal dialogue using natural thought expressions such as 'let me think', 'wait', 'Hmm', 'oh, I see', 'let's break it down', etc, or other natural language thought expressions. After examining the key visual clues, continue with deeper reasoning that connects your observations to the answer. Self-reflection or verification in your reasoning process is encouraged when necessary, though if the answer is straightforward, you may proceed directly to the conclusion. Finally, conclude by placing your final answer in <answer> </answer> tags.

In the video, it is around 5:00 am in China, what is the time in Dubai?

A. around 3am    B. around 11pm    C. around 2am    D. Around 1am    E. around 12am    F. around midnight

Please analyze the video carefully by identifying key segments and their important visual clues within<time> </time>, <caption> </caption>, <think> </think> tags.

Then conduct deep analysis and reasoning to arrive at your answer to the question.

Finally, provide only the single option letter (e.g., A, B, C, D, E, F etc.) within the <answer> </answer> tags. Follow the format specified in the instructions.

**Video-R1-7B**: <think>Let me think about this for a moment. The question is asking about the time in Dubai when it is 5:00 am in China. To answer this, I need to consider the time difference between China and Dubai.

First, let's recall the time zones. China is in the UTC+8 time zone, while Dubai is in the UTC+4 time zone. This means that when it is 5:00 am in China, it is 1 hour earlier in Dubai.

So, if it is 5:00 am in China, it would be 4:00 am in Dubai. Now, I need to match this time with the given options.

Option A: around 3am - This is not correct because it is 1 hour earlier than 4:00 am. Option B: around 11pm - This is not correct because it is 7 hours earlier than 4:00 am. Option C: around 2am - This is not correct because it is 2 hours earlier than 4:00 am. Option D: Around 1am - This is not correct because it is 3 hours earlier than 4:00 am. Option E: around 12am - This is not correct because it is 4 hours earlier than 4:00 am. Option F: around midnight - This is not correct because it is 5 hours earlier than 4:00 am.

After reviewing all the options, the closest match to 4:00 am in Dubai is around 3am in the given options. Therefore, the correct answer is A.</think>
<answer>A</answer>

Figure 14: An example demonstrates Video-R1-7B's inability to follow instructions for generating temporal grounding content within `<time></time>` tags, thereby illustrating the rationale behind the statement in Section 4.3: "Note that Video-R1 is excluded from this evaluation due to its inability to follow our prompt to generate temporal annotations within our templates.".

Figure 15: An example demonstrates that Chain-of-Thought (CoT) information from external tools may mislead Qwen2.5-VL-7B-Instruct, as the CoT reasoning chain could become discontinuous.

