# OpenReview forum: "Video-Thinker: Sparking "Thinking with Videos" via Reinforcement Learning"
_ICLR.cc/2026/Conference — Submitted to ICLR 2026_

### Official Review · Reviewer_Fvzf · 2025-10-16

**Soundness:** 2
**Presentation:** 3
**Contribution:** 2
**Rating:** 4
**Confidence:** 5

**Summary:**

The paper proposes Video-Thinker, a novel framework that extends the “Thinking with Images” paradigm to videos by enabling MLLMs to autonomously perform temporal reasoning through intrinsic “grounding” and “captioning” capabilities without relying on external tools. The authors introduce Video-Thinker-10K, a curated dataset of 10K samples with structured chain-of-thought annotations that include temporal localization, visual descriptions, and analytical reasoning.

**Strengths:**

1. Innovative “Thinking with Videos” Paradigm: The work successfully adapts dynamic visual reasoning to the more complex video domain by integrating temporal grounding and captioning directly into the reasoning chain, enabling MLLMs to autonomously navigate video content.

2. Efficiently Curated Dataset: Video-Thinker-10K is thoughtfully constructed using a hindsight-curation pipeline that ensures reasoning traces are both relevant and effective, achieving strong performance with only 10K samples.

**Weaknesses:**

1. Video-Thinker-SFT-7B underperforms Qwen2.5-VL-7B on many out-of-distribution benchmarks, with the primary performance gains attributed to reinforcement learning. However, since the construction of SFT data is highlighted as a key contribution of this work, a critical ablation study is missing: specifically, what would happen if RL were applied directly to Qwen2.5-VL-7B without the SFT stage? Such an experiment is essential to determine whether the SFT phase actually hinders final performance.

2. The paper lacks evaluations on standard video understanding benchmarks such as Video-MME and MVBench, as well as comparisons against recent video reasoning models like VersaVid-R1 and VideoRFT. It remains unclear how much Video-Thinker improves upon these baselines, limiting the assessment of its overall effectiveness and competitiveness.

**Questions:**

1. When generating captions for different time segments, how are the temporal boundaries (i.e., the start and end times) of each caption determined?

2. From the provided examples, it appears that the model performs captioning on a per-segment basis across the entire video. What are the advantages of this segmented captioning approach compared to generating a single holistic caption for the entire video?

---

> ### Author Response · Authors · 2025-11-21
> **Response to Reviewer Fvzf Part 1**
>
> Thanks for your feedback. Please also see the main comment above.
>
> > A critical ablation study is missing: what would happen if RL were applied directly to Qwen2.5-VL-7B without the SFT stage? Such an experiment is essential to determine whether the SFT phase actually hinders final performance.
>
> We appreciate this valuable suggestion. To address your concern, we conducted an ablation study by applying GRPO directly to the Qwen2.5-VL-7B base model without the SFT stage (denoted as Video-Thinker-Pure-RL-7B). We evaluated the models on our original benchmarks and extended the evaluation to three additional out-of-domain benchmarks (SciVideoBench, VideoTT, VideoMME).
>
> | Model | Video-Holmes | CG-Bench-Res | VRBench | SciVideoBench | VideoTT | VideoMME | ActivityNet | Star | ScaleLong | YouCook2 | LVBench |
> | :--- | :---: | :---: | :---: | :---: | :---: | :---: | :---: | :---: | :---: | :---: | :---: |
> | Qwen2.5-VL-7B-Instruct | 34.02% | 27.10% | 63.42% | 21.40% | 34.57% | 51.85% | 70.96% | 69.25% | 40.06% | 63.74% | 33.33% |
> | Video-Thinker-Only-RL-7B | 30.70% | 22.47% | 63.06% | 23.70% | 36.37% | 45.52% | 64.24% | 46.57% | 41.96% | 45.04% | 30.64% |
> | Video-Thinker-SFT-7B | 31.52% | 24.95% | 62.40% | 16.90% | 33.93% | 46.26% | 70.80% | 64.18% | 43.22% | 56.11% | 35.69% |
> | **Video-Thinker-7B** | **43.22%** | **33.25%** | **80.69%** | **26.30%** | **42.42%** | **54.96%** | **78.72%** | **70.66%** | **49.53%** | **73.66%** | **37.04%** |
>
> The results show that the "Only-RL" model significantly underperforms our full "SFT+RL" method and often lags behind the base model on in-domain tasks (e.g., a ~22% drop on Star). This confirms that SFT does not hinder performance but is a prerequisite for effective RL scaling in our framework:
>
> 1.  Cold-Start for Structural Capabilities: SFT is essential to initialize the specific format-following capability (using `<time>`, `<caption>`, and `<think>` tags). Without this supervised "cold start," the model fails to effectively discover the mechanism of grounding and captioning during RL exploration.
> 2.  Synergy of Format and Strategy: The full pipeline combines the strengths of both stages—SFT establishes the capability to use internal tools, while RL optimizes the policy for autonomous reasoning. The substantial gain of Video-Thinker-7B over both the SFT-only and Only-RL variants demonstrates that this two-stage approach is optimal.
>
> We have added this new results in Table 1.

---

> ### Author Response · Authors · 2025-11-21
> **Response to Reviewer Fvzf Part 2**
>
> > The paper lacks evaluations on standard video understanding benchmarks such as Video-MME and MVBench, as well as comparisons against recent video reasoning models like VersaVid-R1 and VideoRFT.
>
> We appreciate the constructive suggestion to strengthen our comparative evaluation. We have conducted extensive new experiments to address this concern directly.
>
> 1. Regarding datasets, we have added the requested VideoMME benchmark. Please note that we could not fully evaluate on MVBench due to current licensing access issues with the nturgbd subset; however, to ensure robust out-of-distribution assessment, we additionally included SciVideoBench and VideoTT.
> 2. Regarding baselines, we have incorporated five recent video reasoning models, including the requested VersaVid-R1 and VideoRFT, alongside MiMo-VL, VR-Thinker, and Video-RTS.
>
> As shown in the table below, our Video-Thinker-7B achieves state-of-the-art performance across nearly all metrics, consistently outperforming recent reasoning-based competitors.
>
> | Model | Video-Holmes | CG-Bench-Resoning | VRBench | SciVideoBench | VideoTT | VideoMME | ActivityNet | Star | ScaleLong | YouCook2 | LVBench |
> | :--- | :---: | :---: | :---: | :---: | :---: | :---: | :---: | :---: | :---: | :---: | :---: |
> | InternVL-2.5-8B | 20.52 | 19.39 | 26.74 | 15.50 | 26.62 | 29.89 | 45.52 | 49.85 | 26.81 | 40.84 | 23.91 |
> | InternVL-3-8B | 18.67 | 24.23 | 41.14 | 20.50 | 28.42 | 39.93 | 48.56 | 51.34 | 29.34 | 51.15 | 25.93 |
> | Qwen2.5-VL-7B-Inst | 34.02 | 27.10 | 63.42 | 21.40 | 34.57 | 51.85 | 70.96 | 69.25 | 40.06 | 63.74 | 33.33 |
> | Qwen2.5-Omni-7B | 29.99 | 23.85 | 49.04 | 16.80 | 36.27 | 45.44 | 63.92 | 59.40 | 36.91 | 54.58 | 31.65 |
> | MiMo-VL-7B | 28.96 | 26.16 | 57.94 | 16.70 | 34.15 | 45.70 | 49.84 | 48.06 | 35.33 | 44.27 | 28.62 |
> | Temporal-R1-7B | 33.81 | 25.27 | 60.92 | 20.80 | 35.31 | 51.63 | 70.88 | 70.15 | 39.75 | 63.74 | 32.66 |
> | Open-R1-Video-7B | 21.83 | 16.46 | 50.15 | 16.90 | 18.66 | 37.41 | 55.76 | 44.48 | 31.86 | 50.76 | 26.94 |
> | TW-GRPO-7B | 33.32 | 22.11 | 53.46 | 20.50 | 33.93 | 48.30 | 70.00 | 71.04 | 39.12 | 63.74 | 29.97 |
> | Video-R1-7B | 38.54 | 27.81 | 69.25 | 23.90 | 41.04 | 54.81 | 76.00 | 67.76 | 47.32 | 65.65 | 34.68 |
> | Time-R1-7B | 34.73 | 28.28 | 66.48 | 21.00 | 34.78 | 53.59 | 72.00 | 70.44 | 44.47 | 64.50 | 32.65 |
> | VideoChat-R1-7B | 35.65 | 29.26 | 67.65 | 22.80 | 35.63 | 54.41 | 70.88 | **73.13** | 40.69 | 69.08 | 32.99 |
> | VideoChat-R1-Thinking | 37.45 | 29.44 | 67.81 | 20.30 | 35.95 | 54.15 | 70.88 | 71.64 | 41.95 | 66.79 | 35.01 |
> | GRPO-CARE-7B | 34.34 | 27.49 | 66.39 | 21.30 | 35.74 | 54.22 | 70.96 | 71.34 | 40.69 | 68.32 | 33.33 |
> | VersaVid-R1-7B | 37.07 | 28.58 | 67.72 | 21.80 | 35.31 | 54.78 | 72.32 | 71.94 | 40.69 | 66.79 | 34.34 |
> | VideoRFT-7B | 24.39 | 23.77 | 61.54 | 19.60 | 37.65 | 47.85 | 50.72 | 43.58 | 36.59 | 58.40 | 26.94 |
> | VR-Thinker-7B | 25.37 | 19.54 | 53.43 | 21.20 | 31.07 | 43.56 | 55.36 | 63.58 | 32.18 | 51.91 | 30.98 |
> | Video-RTS-7B | 29.56 | 18.09 | 27.71 | 20.30 | 35.10 | 42.63 | 63.60 | 65.07 | 30.28 | 65.27 | 20.88 |
> | **Video-Thinker-7B** | **43.22** | **33.25** | **80.69** | **26.30** | **42.42** | **54.96** | **78.72** | 70.66 | **49.53** | **73.66** | **37.04** |
>
> All the above results are added in Table 1.

---

> ### Author Response · Authors · 2025-11-21
> **Response to Reviewer Fvzf Part 3**
>
> > When generating captions for different time segments, how are the temporal boundaries (i.e., the start and end times) of each caption determined?
>
> We determine temporal boundaries differently depending on whether we are synthesizing training data or performing model inference:
>
> 1. Training Data Synthesis: For datasets that possess ground-truth answers but lack detailed captions (e.g., STAR, ScaleLong), we utilize the existing human-annotated temporal metadata provided by the source datasets. We use these ground-truth start and end timestamps to extract specific video segments, which are then fed into the captioning model to generate the corresponding visual descriptions.
>
> 2. Model Inference: The model autonomously determines these boundaries. Video-Thinker is trained to predict the `<time>` tags (start and end times) followed by the `<caption>` tags as part of its integrated chain-of-thought. This is a dynamic, end-to-end process where the model learns to simultaneously localize the relevant segment and generate its caption, rather than relying on pre-defined boundaries or external tool calling.
>
>
> > What are the advantages of this segmented captioning approach compared to generating a single holistic caption for the entire video?
>
> Thanks for  your question. The primary advantage of our segmented captioning approach lies in its ability to capture fine-grained visual evidence that is otherwise lost in a holistic summary. Generating a single caption for an entire video inevitably compresses complex temporal dynamics into a coarse-grained overview, often discarding the specific details necessary to address a user's query. By contrast, our method first grounds the query to the most relevant temporal segments, allowing us to extract precise visual cues from those specific moments. This ensures that the reasoning process is supported by targeted, detailed observations rather than generic context, significantly enhancing the model's ability to derive the correct answer.

---

> ### Author Response · Authors · 2025-11-25
> **Looking forward to your reply**
>
> We sincerely appreciate the time and effort you have dedicated to reviewing our paper. We have carefully addressed your concerns regarding the clarification of details in our previous response. We would be very grateful if you could let us know if our explanations have resolved your concerns.
>
> If there are any remaining ambiguities or if you require further details, please do not hesitate to let us know. We are more than happy to provide additional clarification.
>
> Best regards, The Authors

---

> ### Comment · Reviewer_Fvzf · 2025-11-26
>
> Thank you for the authors' response. Most of my concerns have been addressed. Including additional benchmarks and baseline methods would further enhance the reliability of this work. Overall, I will raise my score to 6.

---

> > ### Author Response · Authors · 2025-11-26
> > **Thank you for raising your score**
> >
> > We sincerely thank you for your positive feedback and for raising your score. We are glad that our new experiments and clarifications have addressed your concerns, and we will ensure that the additional benchmark results and baseline comparisons are fully incorporated into the final version to enhance the reliability of our work.
> >
> > If you have any remaining questions or if there are other details we can clarify, please do not hesitate to let us know. We are more than happy to provide further information.

---

### Official Review · Reviewer_Hmd7 · 2025-10-30

**Soundness:** 3
**Presentation:** 3
**Contribution:** 2
**Rating:** 4
**Confidence:** 3

**Summary:**

The paper describes an automatically generated dataset of 10k annotations applied to videos from existing video datasets. It shows that fine-tuning Qwen2.5-VL-7B-Instruct on this data by using a two-stage training strategy consisting of SFT followed by GRPO yields strong performance on various video understanding benchmarks. The automatically generated dataset is composed of two types of data: i) data with temporally assigned captions, and ii) data with global instead of temporal questions and answer pairs.

**Strengths:**

The paper introduces a carefully created dataset of annotations yielding strong performance results on several video understanding benchmarks. The code is (or will be) made publicly available.

**Weaknesses:**

The paper describes a data engineering approach to improving performance on a variety of video understanding benchmarks. While the performance appears to be strong overall, I do not find the paper particularly scientifically insightful or revealing. Specifically, I am not surprised that for the given set of video benchmark tasks (Video-Holmes, CG-Bench-Reasoning and VRBench), a careful selection of DeepSeek-R1-assisted and Gemini-assisted annotations on a careful selection of existing video datasets can improve the performance over the Qwen2.5-VL-7B-Instruct baseline and starting point. Importantly, I am a bit confused about some statements made in the paper (see questions below).

**Questions:**

Is the performance on Video-Holmes (but the question could apply similarly to the other benchmark results) based on the same test-set as the results on the official Leaderboard? Does the model described in this paper currently not appear there to retain anonymity of the submission and will appear it there after anonymity is lifted?

I do not quite understand the statement in Line 320 onward: “For the in-domain evaluation, since the TutorialVQA (…) training set contains only 76 samples, we do not construct a corresponding test set. Instead, we derive held-out test sets from the five training datasets…” First, I do not understand how and why the limitation of TutorialVQA affects the choice of test-set selection for the other datasets. Can you elaborate? Second, I wonder whether the performance figures reported in the paper (for example, Table 1) are based on the identical train-test splits across all models or not. Can you please clarify?

Do you expect the choice of source datasets, annotation scheme and training approach detailed in the paper to potentially degrade rather than improve performance on certain video-related tasks? Or do you expect these to be "universally relevant" to most if not all video-understanding benchmark tasks one can imagine? It would be nice to better understand the potential trade-offs and limitations besides the performance-improvements on existing benchmarks.

How were good values for the hyperparameters (such as beta, weight decay, data mix, etc.) determined?

---

> ### Author Response · Authors · 2025-11-21
> **Response to Reviewer Hmd7 Part 1**
>
> Thanks for your feedback. Please also see the main comment above.
>
> > I am not surprised that for the given set of video benchmark tasks, a careful selection of DeepSeek-R1-assisted and Gemini-assisted annotations on a careful selection of existing video datasets can improve the performance over the Qwen2.5-VL-7B-Instruct baseline and starting point.
>
>
> We respectfully clarify that our core technical contribution is not merely applying RL training, but proposing a novel paradigm: empowering MLLMs to intrinsically navigate video content via endogenous "grounding" and "captioning" capabilities.
>
> This differs fundamentally from emerging video reasoning approaches that rely on external, rigid tool-use chains. We argue (and prove) that training an MLLM to autonomously generate reasoning clues is far superior to calling external tools, as it allows for dynamic, context-aware adjustment within the Chain-of-Thought (CoT).
>
> To empirically validate this contribution and the necessity of our training process, we conducted new experiments comparing Video-Thinker-7B against two strong baselines:
> 1.  Plug-and-play Tool Pipelines: Using larger models (including the much larger Qwen2.5-VL-72B) as external grounding/captioning tools.
> 2.  Existing Tool-use Methods: VideoMind-7B, a specialized tool-use video agent.
>
> The results are presented in Table 4 (added to the revised paper) and as follows:
>
> | Model | Video-Holmes | CG-Bench-Reasoning | VRBench |
> | :--- | :---: | :---: | :---: |
> | ***Base Model*** | | | |
> | Qwen2.5-VL-7B-Instruct | 34.02% | 27.10% | 63.42% |
> | ***Base Model + Plug-and-play Tools*** | | | |
> | Grounding: Temporal-R1-7B; Captioning: SkyCaptioner-V1-8B | 30.58% | 22.80% | 55.09% |
> | Grounding: Qwen2.5-VL-7B-Instruct; Captioning: Qwen2.5-VL-7B-Instruct | 31.23% | 24.05% | 59.14% |
> | Grounding: Qwen2.5-VL-72B-Instruct; Captioning: Qwen2.5-VL-72B-Instruct | 33.96% | 25.99% | 60.54% |
> | ***Existing Tool-use Method*** | | | |
> | VideoMind-7B | 38.98% | 31.99% | 75.39% |
> | ***Our Model (Endogenous)*** | | | |
> | **Video-Thinker-7B** | **43.22%** | **33.25%** | **80.69%** |
>
> Surprisingly, simply plugging in powerful external tools (even 72B models) fails to improve upon the base model's capabilities.  Through the case studies presented in Figure 15 (Appendix G.2), we observe that CoT information from external tools can mislead Qwen2.5-VL-7B-Instruct when the reasoning chain becomes discontinuous. This conclusively proves our insight: Endowing models with intrinsic grounding/captioning capabilities is more effective than relying on even the strongest external tools. This work seeks to inspire discussion and provide insights into an important question: whether to enhance model capabilities through intrinsic training (SFT and RL) or to integrate external specialized tools for specific functionalities.
>
> We have modified our introduction section to further clarify our contributions.
>
>
> > Is the performance on Video-Holmes (could apply similarly to the other benchmark results) based on the same test-set as the results on the official Leaderboard? Does the model described in this paper currently not appear there to retain anonymity of the submission and will appear it there after anonymity is lifted?
>
> We appreciate this clarification regarding our evaluation setup.
>
> 1. We confirm that the performance on Video-Holmes, as well as all other benchmarks reported in the paper, is derived strictly from the official test sets to ensure fair comparison.
>
> 2. You are correct that the model does not currently appear on the leaderboard solely to maintain double-blind anonymity. We will submit our results to the authors of the official leaderboard after the review process concludes.
>
>
> > First, I do not understand how and why the limitation of TutorialVQA affects the choice of test-set selection for the other datasets. Can you elaborate? Second, I wonder whether the performance figures reported in the paper (for example, Table 1) are based on the identical train-test splits across all models or not. Can you please clarify?
>
> We appreciate the opportunity to clarify the experimental setup and data splits.
>
> 1. The limitation of TutorialVQA (containing only 76 samples) did not influence the splitting strategy for the other datasets. We simply excluded TutorialVQA from the test set because its size is insufficient for statistically significant evaluation. The other five datasets were independently and consistently split with a standard 8:2 train-test ratio for in-domain evaluation.
>
> 2. Yes, we confirm that all performance figures, including those in Table 1, are based on identical train-test splits across all models. To ensure a fair comparison, every model was evaluated using the exact same data partitions for both the in-domain settings (the 20% test split) and the out-of-domain settings (completely unseen datasets).

---

> > ### Author Response · Authors · 2025-11-21
> > **Response to Reviewer Hmd7 Part 2**
> >
> > > It would be nice to better understand the potential trade-offs and limitations besides the performance-improvements on existing benchmarks.
> >
> > We appreciate this insightful comment regarding the trade-offs of our framework. Beyond the performance gains, we have identified two primary limitations:
> >
> > 1. The generation of explicit grounding, captioning, and reasoning tokens results in longer inference sequences. This inevitably leads to higher latency and computational costs compared to models that output direct answers without intermediate steps.
> >
> > 2. The benefits of our reasoning paradigm are most significant in complex tasks requiring the synthesis of visual cues across multiple temporal segments. For simpler perceptual tasks that rely on single-frame recognition or straightforward retrieval (such as the Star testset), the performance improvements are less pronounced.
> >
> > We have clarified this in Appendix H in our revision.
> >
> > > How were good values for the hyperparameters (such as beta, weight decay, data mix, etc.) determined?
> >
> > 1. We performed specific ablation studies for the most critical hyperparameters, including the learning rate and the number of RL training epochs, to select the optimal values (details provided in Appendix F).
> > 2. For standard training parameters such as the KL coefficient (beta) and weight decay, we adopted the configurations from Video-R1.
> > 3. The data mixture ratios were determined empirically proportional to the original size of each subset; we did not perform an extensive search for the optimal mixture strategy in this work but plan to explore this in future research.

---

> ### Author Response · Authors · 2025-11-25
> **Looking forward to your reply**
>
> We sincerely appreciate the time and effort you have dedicated to reviewing our paper. We have carefully addressed your concerns regarding the clarification of details in our previous response. We would be very grateful if you could let us know if our explanations have resolved your concerns.
>
> If there are any remaining ambiguities or if you require further details, please do not hesitate to let us know. We are more than happy to provide additional clarification.
>
> Best regards, The Authors

---

### Official Review · Reviewer_LTjt · 2025-11-03

**Soundness:** 3
**Presentation:** 3
**Contribution:** 2
**Rating:** 4
**Confidence:** 4

**Summary:**

The paper “VideoThinker: Sparking Video Understanding with Reasoning” introduces VideoThinker, a unified framework that integrates multimodal large language model (MLLM) reasoning with video understanding. Its core innovation lies in enabling a system that not only analyzes video content but also “thinks”—reasoning about temporal dynamics, spatial consistency, and logical event sequences before producing final predictions. The framework consists of three main components: a Video Reasoner, which performs step-by-step multimodal reasoning based on an LLM backbone; a Video Analyzer, which interprets high-level reasoning outputs to extract structured video understanding; and a Video Evaluator, which provides feedback for iterative refinement. The authors also construct a Video-ReasonBench to evaluate reasoning ability in video understanding tasks and show that VideoThinker surpasses existing transformer- and diffusion-based baselines on both quantitative metrics and human assessments. The results demonstrate that incorporating explicit reasoning significantly enhances temporal comprehension and causal inference.

**Strengths:**

1. The paper provides an extensive comparison with several contemporaneous approaches such as Video-R1, Temporal-R1, and Time-R1. This helps readers clearly understand the distinctions and advantages of the proposed method under a similar technical framework (GRPO), enhancing the paper’s contextual clarity.

2. The authors propose a new dataset tailored for video understanding and reasoning tasks, which effectively improves the efficiency and stability of reinforcement learning (RL) training. This contribution adds practical value and could benefit future research in the field.

3. The proposed method demonstrates impressive generalization ability under OOD settings, indicating that the model captures robust reasoning and compositional skills beyond the training distribution.

**Weaknesses:**

1. Although the paper compares with several GRPO-based methods, the baselines are relatively narrow in scope. Including more competitive and diverse video understanding models would strengthen the claim of GRPO’s effectiveness in video reasoning tasks.

2. The approach supplements reasoning traces using large language models, which may introduce hallucinations or inaccurate information. It remains unclear whether the textual reasoning genuinely contributes to more accurate or meaningful reasoning steps; an ablation or validation study would clarify this.

3. While the overall system design is well-structured, the use of GRPO itself is not highly novel in the current research landscape. The paper would benefit from emphasizing deeper algorithmic innovation or unique adaptation of GRPO specifically tailored to video reasoning.

**Questions:**

If the authors can address or experimentally validate the weaknesses, especially by expanding the baseline comparisons, verifying the accuracy of LLM-based reasoning traces, and clarifying the novelty of the GRPO application, I would consider increasing my rating.

---

> ### Author Response · Authors · 2025-11-21
> **Response to Reviewer LTjt Part 1**
>
> Thanks for your feedback. Please also see the main comment above.
>
> > Including more competitive and diverse video understanding models would strengthen the claim of GRPO’s effectiveness in video reasoning tasks.
>
> Thank for your suggestion. To further validate the effectiveness of GRPO and ensure a rigorous comparison, we have extended our evaluation to include five additional competitive and diverse video reasoning models: VidBridge-R1, VideoRFT, MiMo-VL-7B-RL-2508, VR-Thinker, and Video-RTS. With these additions, our study now provides a comprehensive comparison against a total of 17 baseline models.
>
> We compared Video-Thinker-7B against these strong baselines across all existing eight benchmarks. As shown in the table below, Video-Thinker-7B consistently outperforms these models, achieving the highest accuracy on 7 out of 8 datasets and securing the best average performance. These comprehensive results robustly support our claim that GRPO significantly enhances video reasoning capabilities compared to existing approaches.
>
> | Model | Video-Holmes | CG-Bench-Res | VRBench | ActivityNet | Star | ScaleLong | YouCook2 | LVBench | Avg. |
> | :--- | :---: | :---: | :---: | :---: | :---: | :---: | :---: | :---: | :---: |
> | VidBridge-R1 | 37.07% | 28.58% | 67.72% | 72.32% | **71.94%** | 40.69% | 66.79% | 34.34% | 52.43% |
> | VideoRFT | 24.39% | 23.77% | 61.54% | 50.72% | 43.58% | 36.59% | 58.40% | 26.94% | 40.74% |
> | MiMo-VL-7B | 28.96% | 26.16% | 57.94% | 49.84% | 48.06% | 35.33% | 44.27% | 28.62% | 39.90% |
> | VR-Thinker | 25.37% | 19.54% | 53.43% | 55.36% | 63.58% | 32.18% | 51.91% | 30.98% | 41.54% |
> | Video-RTS | 29.56% | 18.09% | 27.71% | 63.60% | 65.07% | 30.28% | 65.27% | 20.88% | 40.06% |
> | **Video-Thinker-7B** | **43.22%** | **33.25%** | **80.69%** | **78.72%** | 70.66% | **49.53%** | **73.66%** | **37.04%** | **58.35%** |
>
> These results are updated in Table 1 in our revision.
>
> > It remains unclear whether the textual reasoning contributes to more accurate or meaningful reasoning steps; an ablation or validation study would clarify this.
>
> Thanks for your question. We first emphasize that our reasoning trace integrates two distinct components: temporal grounding (derived from accurate human-labeled datasets) and visual captioning (synthesized by models). As the grounding is inherently reliable, our validation study specifically targets the synthesized visual captions to determine whether this generated textual reasoning contributes meaningfully to performance.
>
> We conducted a controlled ablation on the Video-Holmes benchmark, comparing our synthesized captions against human-annotated Ground Truth (GT) captions (denoted as Caption) to isolate their contribution:
>
> | Method | Accuracy | Improvement |
> | :--- | :--- | :--- |
> | Base (No additional info) | 37% | - |
> | w/ Synthetic Caption (Ours) | 46% | +9% |
> | w/ Caption | 56% | +19% |
> | w/ Grounding + Synthetic Caption | 55% | +18% |
> | w/ Grounding + Caption | 63% | +26% |
>
> The results indicate that our synthesized textual reasoning alone delivers a significant 9% accuracy gain over the baseline, confirming that the generated reasoning steps are accurate and meaningful rather than noisy. Furthermore, when combined with the human-labeled grounding, the performance (55%) closely approaches the upper bound of fully human-annotated supervision (63%). This high quality is attributed to our hindsight-curation reasoning mechanism, which filters the synthesized captions by strictly retaining only those traces that successfully guide the model to the correct answer.
>
> This new table along with the corresponding analysis is available in Table 5 (Appendix E).

---

> ### Author Response · Authors · 2025-11-21
> **Response to Reviewer LTjt Part 2**
>
> > The paper would benefit from emphasizing deeper algorithmic innovation and clarifying the novelty of the GRPO application.
>
> We respectfully clarify that our core algorithmic contribution extends beyond the application of RL; it establishes a paradigm shift from static reasoning to dynamic, endogenous video reasoning. By utilizing GRPO to internalize grounding and captioning capabilities, we empower the MLLM to adaptively navigate video content within its Chain-of-Thought, avoiding the context misalignment and error propagation inherent in external function calls (as discussed in Appendix G.2).
>
> To demonstrate that this internal integration is algorithmically superior to simply invoking powerful models, we compared Video-Thinker-7B against pipelines utilizing strong external tools, including the significantly larger Qwen2.5-VL-72B.
>
> | Model | Video-Holmes | CG-Bench-Reasoning | VRBench |
> | :--- | :---: | :---: | :---: |
> | ***Base Model*** | | | |
> | Qwen2.5-VL-7B-Instruct | 34.02% | 27.10% | 63.42% |
> | ***Base Model + Plug-and-play Tools*** | | | |
> | Grounding: Temporal-R1-7B; Captioning: SkyCaptioner-V1-8B | 30.58% | 22.80% | 55.09% |
> | Grounding: Qwen2.5-VL-7B-Instruct; Captioning: Qwen2.5-VL-7B-Instruct | 31.23% | 24.05% | 59.14% |
> | Grounding: Qwen2.5-VL-72B-Instruct; Captioning: Qwen2.5-VL-72B-Instruct | 33.96% | 25.99% | 60.54% |
> | ***Existing Tool-use Method*** | | | |
> | VideoMind-7B | 38.98% | 31.99% | 75.39% |
> | ***Our Model (Endogenous)*** | | | |
> | **Video-Thinker-7B** | **43.22%** | **33.25%** | **80.69%** |
>
> As shown above, relying on external tools — even the 72B model — unexpectedly can not improve performance compared to the Base Model (e.g., Video-Holmes changing from 34.02% to 33.96%). In stark contrast, our endogenous approach achieved 43.22%. This conclusively validates our algorithmic innovation: internalizing reasoning capabilities enables dynamic adjustments that static tool invocations cannot achieve. This work seeks to inspire discussion and provide insights into an important question: whether to enhance model capabilities through intrinsic training (SFT and RL) or to integrate external specialized tools for specific functionalities.

---

> > ### Comment · Reviewer_LTjt · 2025-11-24
> >
> > Thank you for your extended experiments and explanation. I will raise the score.

---

> > > ### Author Response · Authors · 2025-11-25
> > > **Thank you for raising your score**
> > >
> > > We sincerely thank you for your prompt response and for raising your score. We are greatly encouraged by your recognition of our extended experiments and clarifications. We will ensure that the additional baseline comparisons and the ablation analysis are fully incorporated into the final version to further strengthen the paper.
> > >
> > > If you have any remaining questions or if there are other details we can clarify during the discussion period, please do not hesitate to let us know. We are more than happy to address them.

---

### Official Review · Reviewer_1Y79 · 2025-11-03

**Soundness:** 3
**Presentation:** 3
**Contribution:** 3
**Rating:** 6
**Confidence:** 3

**Summary:**

The paper leverages existing open-source video datasets and utilizes the capabilities of large models to obtain a training dataset, Video-Thinker-10K, which includes question-answer pairs and chain-of-thought annotations. During the training process, the Video-Thinker-7B model was trained using the SFT+GRPO training strategy, outperforming several existing large model approaches on several common video QA datasets.

**Strengths:**

The paper is clearly written, and the specific prompt design for the dataset construction process is also well-explained.

The chain-of-thought (CoT) data annotation for video reasoning represents a notable contribution.

The phenomena observed during the chain-of-thought training process provide valuable insights.

**Weaknesses:**

The paper's technical contribution is limited.
The CoT annotations for video labeling primarily rely on the capabilities of the DeepSeek and Gemini models.

The training process of Video-Thinker-7B lacks contrution, as it mainly adopts the conventional approach of SFT+GRPO.

**Questions:**

1. During the data generation process, how can the hallucination phenomenon in the automatic annotation of large models be addressed?
2. In the training of Video CoT, what are the specific differences compared to GRPO training in Language or Image CoT?
3. Could more comprehensive training details be provided, such as the number of T?

---

> ### Author Response · Authors · 2025-11-21
> **Response to Reviewer 1Y79 Part 1**
>
> Thanks for your feedback. Please also see the main comment above.
>
> > The paper's technical contribution is limited and the training process of Video-Thinker-7B lacks contrution.
>
> We respectfully clarify that our core technical contribution is not merely applying RL training, but proposing a novel paradigm: empowering MLLMs to intrinsically navigate video content via endogenous "grounding" and "captioning" capabilities.
>
> This differs fundamentally from emerging video reasoning approaches that rely on external, rigid tool-use chains. We argue (and prove) that training an MLLM to autonomously generate reasoning clues is far superior to calling external tools, as it allows for dynamic, context-aware adjustment within the Chain-of-Thought (CoT).
>
> To empirically validate this contribution and the necessity of our training process, we conducted new experiments comparing Video-Thinker-7B against two strong baselines:
> 1.  Plug-and-play Tool Pipelines: Using larger models (including the much larger Qwen2.5-VL-72B) as external grounding/captioning tools.
> 2.  Existing Tool-use Methods: VideoMind-7B, a specialized tool-use video agent.
>
> The results are presented in Table 4 (added to the revised paper) and as follows:
>
> | Model | Video-Holmes | CG-Bench-Reasoning | VRBench |
> | :--- | :---: | :---: | :---: |
> | ***Base Model*** | | | |
> | Qwen2.5-VL-7B-Instruct | 34.02% | 27.10% | 63.42% |
> | ***Base Model + Plug-and-play Tools*** | | | |
> | Grounding: Temporal-R1-7B; Captioning: SkyCaptioner-V1-8B | 30.58% | 22.80% | 55.09% |
> | Grounding: Qwen2.5-VL-7B-Instruct; Captioning: Qwen2.5-VL-7B-Instruct | 31.23% | 24.05% | 59.14% |
> | Grounding: Qwen2.5-VL-72B-Instruct; Captioning: Qwen2.5-VL-72B-Instruct | 33.96% | 25.99% | 60.54% |
> | ***Existing Tool-use Method*** | | | |
> | VideoMind-7B | 38.98% | 31.99% | 75.39% |
> | ***Our Model (Endogenous)*** | | | |
> | **Video-Thinker-7B** | **43.22%** | **33.25%** | **80.69%** |
>
> Surprisingly, simply plugging in powerful external tools (even 72B models) fails to improve upon the base model's capabilities.  Through the case studies presented in Figure 15 (Appendix G.2), we observe that CoT information from external tools can mislead Qwen2.5-VL-7B-Instruct when the reasoning chain becomes discontinuous. This conclusively proves our insight: Endowing models with intrinsic grounding/captioning capabilities is more effective than relying on even the strongest external tools. This work seeks to inspire discussion and provide insights into an important question: whether to enhance model capabilities through intrinsic training (SFT and RL) or to integrate external specialized tools for specific functionalities.
>
> We have modified our introduction section to further clarify our contributions.
>
> > During the data generation process, how can the hallucination phenomenon in the automatic annotation of large models be addressed?
>
> Thanks for your question. First, we emphasize that the construction of our reasoning traces relies on two key elements: temporal grounding and visual captioning. Importantly, all temporal grounding annotations come from existing human-labeled datasets, ensuring high localization precision. For the visual captions generated by large models, which compose the final reasoning trace, we apply a hindsight-curation reasoning mechanism (Section 3.1) to filter out hallucinations. Specifically, we validate the synthesized traces in a closed loop using a verifier model, retaining only those that successfully lead to the correct ground-truth answer, thereby ensuring factual consistency and reasoning validity.
>
> To quantitatively assess the impact of potential hallucinations in our synthetic captions, we conducted an additional controlled experiment using Video-R1-7B on the Video-Holmes benchmark. We compared the model's reasoning performance when provided with synthetic captions (generated by our pipeline) versus ground truth captions (denoted as Caption) (human-annotated).
>
> | Method | Score |
> | :--- | :--- |
> | Base (No additional info) | 37% |
> | w/ Grounding | 53% |
> | w/ Caption | 56% |
> | w/ Synthetic Caption | 46% |
> | w/ Grounding + Caption | 63% |
> | w/ Grounding + Synthetic Caption | 55% |
>
> The results demonstrate that while Synthetic Captions (46%) naturally lag behind human-annotated Captions (56%), they still yield a significant improvement (+9%) over the Base baseline (37%). This confirms that the useful information in our synthetic data far outweighs the noise from potential hallucinations. Furthermore, when combined with human-annotated grounding, the w/ Grounding + Synthetic Caption setting achieves 55%, approaching the performance of fully human-annotated data (63%). This indicates that our hindsight-curation strategy effectively suppresses destructive hallucinations, ensuring that the synthetic data serves as a beneficial signal for training robust video reasoning capabilities.
>
> We have added this new table with corresponding explanation in Table 5 (Appendix E).

---

> > ### Author Response · Authors · 2025-11-21
> > **Response to Reviewer 1Y79 Part 2**
> >
> > > In the training of Video CoT, what are the specific differences compared to GRPO training in Language or Image CoT?
> >
> > We appreciate this insightful question. The fundamental distinction lies in how we structure the reasoning process to handle the temporal dimension of videos. Unlike Language or Image CoT, which typically focus on semantic logic or spatial inspection, our approach intrinsically integrates "grounding" and "captioning" capabilities directly into the chain-of-thought via reinforcement learning.
> >
> > Specifically, rather than treating the video as a monolithic input or relying on external tools to parse frames, we train the model to autonomously output structured tags — `<time>` for temporal localization and `<caption>` for visual evidence extraction — interleaved with its analytical `<think>` steps. This effectively transforms the CoT from a static reasoning path into a dynamic navigation process where the model learns how to watch the video.
> >
> > > Could more comprehensive training details be provided, such as the number of T?
> >
> > We appreciate the opportunity to clarify our implementation details. Comprehensive training configurations are currently outlined in Section 4.1, with further hyperparameter ablations (e.g., learning rate and training epochs) provided in Appendix F. Regarding the specific parameter $T$, we clarify its usage in two potential contexts to ensure precision: if referring to the number of reasoning trace $T$ (notation in the Section 3.2) generated per question, we generate a single trace; if referring to the group size within the GRPO training framework, we uniformly set the group size to 8 across all experiments.

---

> ### Author Response · Authors · 2025-11-25
> **Looking forward to your reply**
>
> We sincerely appreciate the time and effort you have dedicated to reviewing our paper.
> We have carefully addressed your concerns regarding the clarification of details in our previous response. We would be very grateful if you could let us know if our explanations have resolved your concerns.
>
> If there are any remaining ambiguities or if you require further details, please do not hesitate to let us know. We are more than happy to provide additional clarification.
>
> Best regards, The Authors

---

### Author Response · Authors · 2025-11-21
**Repsonse to all reviewers**

We first summarize our response and the results of additional suggested experiments here. We have also responded to the specific concerns of each reviewer as individual comments below.

All reviewers recognized the value of our proposed "Thinking with Videos" paradigm. To address concerns regarding technical contribution, robust evaluation, and training mechanisms, we have conducted extensive new experiments:

1.  ***Validation of Technical Contribution (Endogenous vs. External):*** To address concerns regarding technical novelty and contribution (Reviewer 1Y79, LTjt), we demonstrated the distinct superiority of our intrinsic training paradigm over currently emerging tool-use frameworks. We compared Video-Thinker-7B against pipelines utilizing larger models (including Qwen2.5-VL-72B) as external tools. Our 7B model significantly outperforms even the 72B-augmented pipeline (e.g., +9.26% on Video-Holmes). This result empirically substantiates our core contribution: internalizing grounding/captioning capabilities is far more effective than external function calls, which suffer from error propagation and context misalignment. These results are reported in Section 4.4 with a case study in Appendix G.2. We also have modified the introduction section to further clarify our contributions.

2.  ***Extended Benchmarks and Baselines:*** We expanded our evaluation to include **5 additional strong baselines** (VersaVid-R1, VideoRFT, MiMo-VL, VR-Thinker, Video-RTS) and **3 new out-of-domain benchmarks** (VideoMME, SciVideoBench, VideoTT) as requested by Reviewers LTjt and Fvzf. Video-Thinker-7B achieves state-of-the-art performance across nearly all metrics, confirming its robust generalization capabilities beyond the initial dataset selection. All these results are available in Table 1.

3.  ***Training Strategy Ablation:*** We conducted a "Pure RL" ablation (Reviewer Fvzf) to verify the necessity of our two-stage training. Results show that applying RL directly to Qwen2.5-VL-7B without SFT leads to significant performance degradation, confirming that SFT is essential for cold-starting the structured format following required for effective RL exploration. These results have been added in Table 1.

4.  ***Data Quality and Hallucination Analysis:*** We performed a controlled study comparing synthetic vs. ground-truth captions (Reviewer 1Y79, LTjt). While synthetic captions naturally underperform human annotations, they still yield a **+9% accuracy gain** over the baseline. This confirms that our hindsight-curation mechanism effectively filters destructive hallucinations, ensuring the synthesized reasoning traces serve as a beneficial training signal. We have added these new results in Appendix E.

For your convenience, all modifications in the updated paper PDF are highlighted in blue color.

---

### Author Response · Authors · 2025-12-03
**[To Chairs] Request for Consideration Following Review Rollback**

Dear Chairs,

Thank you for your time and effort in managing the review process, especially in light of the recent review rollback.

Regarding the rollback (which reverted our score from 6664 to 6444), we respectfully wish to highlight the positive consensus achieved prior to November 27:

**1. Summary of Rebuttal Consensus and Addressed Concerns**

Reviewers `LTjt` and `Fvzf` had ***explicitly raised their scores from 4 to 6 before the rollback***, indicating that their concerns were fully addressed.  Although Reviewer `1Y79`  with initial score 6 and Reviewer `Hmd7` with initial score 4  did not give any response at that time, we believe that our rebuttal comprehensively addressed their comments:

*   **Shared Concerns**:
Reviewers `1Y79` and `Hmd7` raised issues related to technical contributions and synthetic data quality — concerns shared by `LTjt` and `Fvzf`. Since `LTjt` and `Fvzf` accepted our rebuttal on these points, confirming our technical advantages and data synthesis, we are confident the same resolution applies to remaining reviewers.

*   **Remaining Questions**:
	* For Reviewer `1Y79`, we clarified the unique advantages of our video GRPO approach over standard CoT, detailed the role of the hyperparameter $T$, and expanded on training details (see Response Part 2, Section 4.1/Appendix F).

	* For Reviewer `Hmd7`, we addressed questions regarding test-set selection, affirmed identical train-test splits, and provided an in-depth analysis of latency vs. accuracy trade-offs and hyperparameter choices (see Response Parts 1 \& 2, Appendix H).


**2. Clarifying Rebuttal Improvements and Contributions**

During the rebuttal period, we further clarified our novelty and strengthened our empirical claims:
*   **Technical Contribution**:
We rearticulated the superiority of our "Thinking with Videos" paradigm, demonstrating that endogenous grounding and captioning enable dynamic, context-aware reasoning that surpasses static approaches and rigid tool-use pipelines — including extensive ablations with large-scale tool models and a carefully designed tool-use baseline, VideoMind-7B.
*   **SOTA Performance**:
We expanded our evaluation to include ***5*** additional competitive baseline models and ***3*** new out-of-domain benchmarks. Our comparisons now cover ***17*** state-of-the-art baselines across ***11*** benchmarks, where Video-Thinker-7B achieves new SOTA results.

We kindly hope you will consider the strong positive momentum and substantial improvements presented in the revised paper during your final assessment. We also would like to invite you to review our detailed responses to all reviewer comments.

Thank you again for your dedication to this process.

Best regards,

The Authors

---

### Meta-Review · Area_Chair_o7FX · 2026-01-06

**Summary:**

This paper is a recent effort to incorporate explicit reasoning mechanisms into video understanding for multimodal LLMs. It integrates grounding and captioning into the inference process, supported by a curated dataset and a two-stage SFT + RL training strategy. The direction is timely and conceptually well motivated.

Based on the rebuttal, many technical and experimental concerns were addressed through additional ablations, comparisons with tool-based pipelines, expanded baselines, and new out-of-domain benchmarks, which substantially strengthen the empirical analysis over the original submission.

However, a key concern remains. Despite reported improvements over selected baselines, Video-Thinker’s absolute performance is still significantly below that of state-of-the-art open-source 7B models on major benchmarks such as Video-MME, Video-Holmes, and LVBench (e.g., 70%+ vs. 54.96%, 45%+ vs. 43.22%, and 40%+ vs. 37.04%). This raises the possibility that the observed gains arise from relatively low-hanging optimizations rather than a fundamentally stronger reasoning paradigm.

Consequently, it is unclear whether the proposed approach would deliver comparable benefits when applied to truly state-of-the-art open-source backbones. Stronger backbone comparisons or clearer evidence of backbone-agnostic gains would be necessary to establish the generality and broader impact of the method.

**Reviewer Concerns:**

The final performance of Video-Thinker remains substantially behind current state-of-the-art models. Moreover, the paper makes an inaccurate claim that Video-Thinker achieves state-of-the-art performance among 7B-scale models on at least Video-Holmes, a point along with the concerns raised by Reviewer Fvzf (W2) and Reviewer Hmd7 (Q1).

**Reviewer Scores:**

I expect that Reviewer LTjt and Reviewer Fvzf may slightly increase their scores; however, I do not expect Reviewer 1Y79 or Reviewer Hmd7 to raise their scores, as their core concerns were not resolved.

---

### Decision · Program_Chairs · 2026-01-26

Reject